# On Characterizing the Trade-off in Invariant Representation Learning

**Bashir Sadeghi**                                                          *sadeghib@msu.edu*
**Sepehr Dehdashtian**                                                        *sepehr@msu.edu*
**Vishnu Naresh Boddeti**                                                     *vishnu@msu.edu*
*Department of Computer Science and Engineering*
*Michigan State University*

**Reviewed on OpenReview:** *https://openreview.net/forum?id=3gfpBR1ncr&referrer*

## Abstract

Many applications of representation learning, such as privacy preservation, algorithmic fairness, and domain adaptation, desire explicit control over semantic information being discarded. This goal is formulated as satisfying two objectives: maximizing utility for predicting a target attribute while simultaneously being invariant (independent) to a known semantic attribute. Solutions to invariant representation learning (IRepL) problems lead to a trade-off between utility and invariance when they are competing. While existing works study bounds on this trade-off, two questions remain outstanding: 1) *What is the exact trade-off between utility and invariance?* and 2) *What are the encoders (mapping the data to a representation) that achieve the trade-off, and how can we estimate it from training data?* This paper addresses these questions for IRepLs in reproducing kernel Hilbert spaces (RKHS)s. Under the assumption that the distribution of a low-dimensional projection of high-dimensional data is approximately normal, we derive a closed-form solution for the global optima of the underlying optimization problem for encoders in RKHSs. This yields closed formulae for a near-optimal trade-off, corresponding optimal representation dimensionality, and the corresponding encoder(s). We also numerically quantify the trade-off on representative problems and compare them to those achieved by baseline IRepL algorithms. Code is available at `https://github.com/human-analysis/tradeoff-invariant-representation-learning`.

## 1 Introduction

Real-world applications of representation learning often have to contend with objectives beyond predictive performance. These include cost functions corresponding to invariance (e.g., to photometric or geometric variations), semantic independence (e.g., to age or race for face recognition systems), privacy (e.g., mitigating leakage of sensitive information (Roy & Boddeti, 2019)), algorithmic fairness (e.g., demographic parity (Madras et al., 2018)), and generalization across multiple domains (Ganin et al., 2016), to name a few.

At its core, the goal of the aforementioned formulations of representation learning is to satisfy two competing objectives: Extracting as much information necessary to predict a target label $Y$ (e.g., face identity) while *intentionally* and *permanently* suppressing information about a given semantic attribute $S$ (e.g., age or gender). See Figure 1 (a) for illustration. Let $Z$ be the encoding of the input data from which the target attribute $Y$ can be predicted. When the statistical dependency between $Y$ and $S$ is not negligible, learning a representation $Z$ that is invariant to the semantic attribute $S$ (i.e., $Z \perp\!\!\!\perp S$) will necessarily degrade the performance of the target prediction, i.e., there exists a trade-off between utility and invariance. The existence of a trade-off has been well established, both theoretically and empirically, under various contexts of representation learning such as fairness (Menon & Williamson, 2018; Zhao & Gordon, 2019; Gouic et al., 2020; Zhao, 2021), invariance (Zhao et al., 2020), and domain adaptation (Zhao et al., 2019b). However, much of this body of work only establishes bounds on the trade-off rather than a *precise* characterization. As such, two aspects of the trade-off in invariant representation learning (IRepL) are unknown, including i) *exact* characterization of the trade-off inherent to IRepL and ii) a learning algorithm that achieves the trade-off. Under the

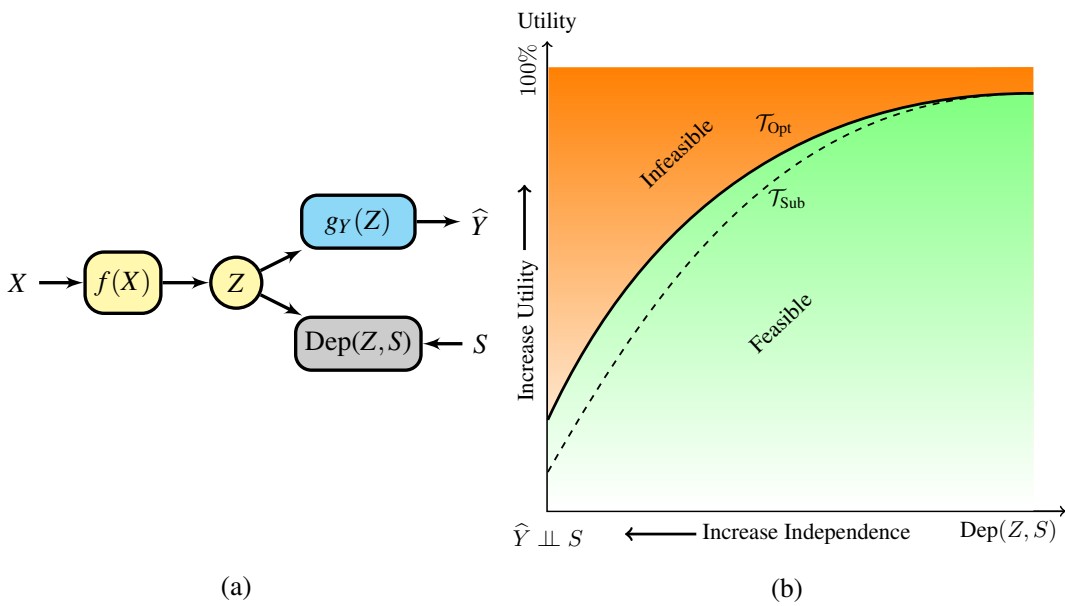

Figure 1: (a): Invariant representation learning seeks a representation $Z = f(X)$ that contains as much information necessary for the downstream target predictor $g_Y$ while being independent of the semantic attribute $S$. (b): The trade-off (denoted by $\mathcal{T}_{\text{Opt}}$) between utility (target task performance) and invariance (measured by the dependence metric $\text{Dep}(Z, S)$) is induced by a controlled representation learner in the hypothesis class of all Borel functions.

assumption that the distribution of a low-dimensional projection of high-dimensional data is approximately normal, this paper studies and establishes the aforementioned properties by constraining function classes to reproducing kernel Hilbert spaces (RKHS)s.

Ideally, the utility-invariance trade-off is defined as a bi-objective optimization problem:

$$\inf_{f \in \mathcal{H}_X, \, g_Y \in \mathcal{H}_Y} \mathbb{E}_{XY} \left[ L_Y \left( g_Y \left( f(X) \right), Y \right) \right] \quad \text{such that} \quad \text{Dep} \left( f(X), S \right) \leq \epsilon, \tag{1}$$

where $f$ is the encoder that extracts the representation $Z = f(X)$ from $X$, $g_Y$ predicts $\widehat{Y}$ from the representation $Z$, $\mathcal{H}_X$ and $\mathcal{H}_Y$ are the corresponding hypothesis classes, and $L_Y$ is the loss function for predicting the target attribute $Y$. The function $\text{Dep}(\cdot, \cdot) \geq 0$ is a parametric or non-parametric measure of statistical dependence, i.e., $\text{Dep}(Q, U) = 0$ implies $Q$ and $U$ are independent, and $\text{Dep}(Q, U) > 0$ implies $Q$ and $U$ are dependent with larger values indicating greater degrees of dependence. The scalar $\epsilon \geq 0$ is a user-defined parameter that controls the trade-off between the two objectives, with $\epsilon \to \infty$ being the standard scenario that has no invariance constraints with respect to (w.r.t.) $S$. In contrast, $\epsilon \to 0$ enforces $Z \perp\!\!\!\perp S$ (i.e., total invariance). Involving all Borel functions in $\mathcal{H}_X$ and $\mathcal{H}_Y$ ensures that the best possible trade-off is included within the feasible solution space. For example, when $\epsilon \to \infty$ and $L_Y$ is MSE loss, the optimal Bayes estimator, $g_Y \left( f(X) \right) = \mathbb{E}\left[ Y \mid X \right]$ is attainable.

In this paper, we consider the linear combination of utility and invariance in (1) and define the optimal utility-invariance trade-off (denoted by $\mathcal{T}_{\text{Opt}}$) as a single objective optimization problem:

**Definition 1.**

$$\mathcal{T}_{\text{Opt}} := \inf_{f \in \mathcal{H}_X} \left\{ (1 - \lambda) \inf_{g_Y \in \mathcal{H}_Y} \mathbb{E}_{X,Y} \left[ L_Y \left( g_Y \left( f(X) \right), Y \right) \right] + \lambda \, \text{Dep} \left( f(X), S \right) \right\}, \quad 0 \leq \lambda < 1, \tag{2}$$

where $\lambda$ controls the trade-off between utility and invariance. For example, $\lambda = 0$ corresponds to ignoring the invariance and only optimizing the utility, while $\lambda \to 1$ corresponds to $Z \perp\!\!\!\perp S$.

The motivations behind considering this single-objective IRepL are (i) any solution to this simplified problem is a solution to the bi-objective problem in (1), (ii) even (2) is challenging to solve, and (iii) existing works have not

thoroughly investigated (2). An illustration of the utility-invariance trade-off is illustrated in Figure 1 (b). In this paper, we restrict $\mathcal{H}_X$ to be some RKHSs and $\mathrm{Dep}(Z, S)$ to be a simplified version of the Hilbert-Schmidt Independence Criterion (HSIC) (Gretton et al., 2005a). Further, we replace the target loss function in (2) by $\mathrm{Dep}(Z, Y)$ as presented and justified in Sections 3.2 and 5.2.

**Summary of Contributions:** i) We design a dependence measure that accounts for all modes of dependence between $Z$ and $S$ [1] (under a mild assumption) while allowing for analytical tractability. ii) We employ functions in RKHSs and obtain closed-form solutions for the IRepL optimization problem. Consequently, we precisely characterize a near-optimal approximation of $\mathcal{T}_{\mathrm{Opt}}$ via encoders restricted to RKHSs. iii) We obtain a closed-form estimator for the encoder that achieves a near-optimal trade-off and establish its numerical convergence. iv) Using random Fourier features (RFF) (Rahimi et al., 2007), we provide a scalable version (in terms of both memory and computation) of our IRepL algorithm. v) We numerically quantify our $\mathcal{T}_{\mathrm{Opt}}$ (denoted by K-$\mathcal{T}_{\mathrm{Opt}}$) on an illustrative problem as well as large-scale real-world datasets, Folktables (Ding et al., 2021) and CelebA (Liu et al., 2015), where we compare K-$\mathcal{T}_{\mathrm{Opt}}$ to those obtained by existing works.

## 2 Related Work

### 2.1 Invariant Representation Learning

The basic idea of representation learning that discards unwanted semantic information has been explored under many contexts like invariant, fair, or privacy-preserving learning. In domain adaptation (Ganin & Lempitsky, 2015; Tzeng et al., 2017; Zhao et al., 2018), the goal is to learn features that are independent of the data domain. In fair learning (Dwork et al., 2012; Ruggieri, 2014; Feldman et al., 2015; Calmon et al., 2017; Zemel et al., 2013; Edwards & Storkey, 2015; Beutel et al., 2017; Xie et al., 2017; Zhang et al., 2018; Song et al., 2019; Madras et al., 2018; Bertran et al., 2019; Creager et al., 2019; Locatello et al., 2019; Mary et al., 2019; Martinez et al., 2020; Sadeghi et al., 2019), the goal is to discard the demographic information that leads to unfair outcomes. Similarly, there is growing interest in mitigating unintended leakage of private information from representations (Hamm, 2017; Coavoux et al., 2018; Roy & Boddeti, 2019; Xiao et al., 2020; Dusmanu et al., 2021).

A vast majority of this body of work is empirical. They implicitly look for single or multiple points on the trade-off between utility and semantic information and do not explicitly seek to characterize the whole trade-off front. Overall, these approaches are not concerned with or aware of the inherent utility-invariance trade-off. In contrast, with the cost of restricting encoders to lie in some RKHSs, we *precisely* characterize the trade-off and propose a practical learning algorithm that achieves this trade-off.

### 2.2 Adversarial Representation Learning

Most practical approaches for learning fair, invariant, domain adaptive, or privacy-preserving representations discussed above are based on adversarial representation learning (ARL). ARL is typically formulated as

$$\inf_{f \in \mathcal{H}_X} \left\{ (1 - \lambda) \inf_{g_Y \in \mathcal{H}_Y} \mathbb{E}_{X,Y} \left[ L_Y \left( g_Y \left( f(X) \right), Y \right) \right] - \lambda \inf_{g_S \in \mathcal{H}_S} \mathbb{E}_{X,S} \left[ L_S \left( g_S \left( f(X) \right), S \right) \right] \right\}, \tag{3}$$

where $L_S$ is the loss function of a hypothetical adversary $g_S$, who intends to extract the semantic attribute $S$ through the best estimator within the hypothesis class $\mathcal{H}_S$, and $0 \leq \lambda < 1$ is the utility-invariance trade-off parameter. ARL is a special case of (2) where the negative loss of the adversary, $- \inf_{g_S \in \mathcal{H}_S} \mathbb{E}_{X,S} \left[ L_S \left( g_S \left( f(X) \right), S \right) \right]$ plays the role of $\mathrm{Dep}(f(X), S)$. However, this form of adversarial learning suffers from a critical drawback. The induced independence measure is not guaranteed to account for all modes of non-linear dependence between $S$ and $Z$ if the adversary loss function $L_S$ is not bounded like MSE or cross-entropy (Adeli et al., 2021; Grari et al., 2020). In the case of MSE loss, even if the loss is maximized at a bounded value, where the corresponding representation $Z = f(X)$ is also bounded, it still does not guarantee that $Z \perp\!\!\!\perp S$ is attainable (see Appendix H for more details). This implies that designing the adversary loss in ARL to account for all types of dependence is challenging and can be infeasible for some loss functions.

---

[1] By "all modes of dependence" we mean all types of linear and non-linear relations, in contrast to only linear or monotonic relations.

### 2.3 Trade-Offs in Invariant Representation Learning:

Prior work has established the existence of trade-offs in IRepL, both empirically and theoretically. In the following, we categorize them based on properties of interest.

**Restricted Class of Attributes:** A majority of existing work considers IRepL trade-offs under restricted settings, e.g., binary and/or categorical attributes $Y$ and $S$. For instance, Zhao et al. (2019a) uses information-theoretic tools and characterizes the utility-fairness trade-off in terms of lower bounds when both $Y$ and $S$ are binary labels. Later McNamara et al. (2019) provided both upper and lower bounds for binary labels. By leveraging Chernoff bound, Dutta et al. (2020) proposed a construction method to generate an ideal representation beyond the input data to achieve perfect fairness while maintaining the best performance on the target task. In the case of categorical features, a lower bound on utility-fairness trade-off has been provided by Zhao & Gordon (2019) for the total invariance scenario (i.e., $Z \perp\!\!\!\perp S$). In contrast to this body of work, our trade-off analysis applies to multi-dimensional continuous/discrete attributes. To our knowledge, the only prior results under a general setting are Sadeghi et al. (2019) and Zhao et al. (2020). However, in Zhao et al. (2020), both $S$ and $Y$ are restricted to be continuous/discrete or binary simultaneously (e.g., it is not possible to have $Y$ binary while $S$ is continuous).

**Characterizing Exact versus Bounds on Trade-Off:** To the best of our knowledge, all existing approaches except Sadeghi et al. (2019), which obtains the trade-off for the linear dependence only, characterize the trade-off in terms of upper and/or lower bounds. In contrast, we *precisely* characterize a near-optimal trade-off with closed-form expressions for encoders belonging to some RKHSs.

**Optimal Encoder and Representation:** Another property of practical interest is the optimal encoder that achieves the desired point on the utility-invariance trade-off and the corresponding representation(s). Existing works which only study bounds on the trade-off do not obtain the encoder that achieves those bounds. For example, Sadeghi et al. (2019) develop a learning algorithm that obtains a globally optimal encoder, but only under a linear dependence measure between $Z$ and $S$. HSIC, a universal measure of dependence, has been adopted by prior work (e.g., Quadrianto et al. (2019)) to quantify all types of dependencies between $Z$ and $S$. However, these methods adopt stochastic gradient descent for optimizing the underlying non-convex optimization problem. As such, they fail to provide guarantees that the representation learning problem converges to a global optimum. In contrast, we obtain a closed-form solution for the globally optimal encoder and its corresponding representation while detecting all modes of dependence between $Z$ and $S$.

### 2.4 Kernel Method

The technical machinery of our kernel method for representation learning is closely related to kernelized component analysis (Schölkopf et al., 1998). Kernel methods have been previously used for fair representation learning by Pérez-Suay et al. (2017) where the Rayleigh quotient is employed to only search for a single point in the utility-invariance trade-off. To find the entire trade-off, Sadeghi et al. (2019) used kernelized ARL with a linear adversary and target estimator. Kernel methods also have been used to measure all modes of dependence between two RVs, pioneered by Bach & Jordan (2002) in kernel canonical correlation (KCC). Building upon KCC, later, Gretton et al. (2005a;b; 2006) have introduced HSIC, constrained covariance (COCO), and maximum mean discrepancy (MMD), to name a few. Inspired by these works, a variation of HSIC is adopted as a measure of dependence in this paper.

## 3 Problem Setting

### 3.1 Notation

Scalars are denoted by regular lowercase letters, e.g., $r$, $\lambda$. Deterministic vectors are denoted by boldface lowercase letters, e.g., $\boldsymbol{x}$, $\boldsymbol{s}$. We denote both scalar-valued and multidimensional random variables (RV)s by regular upper case letters, e.g., $X$, $S$. Deterministic matrices are denoted by boldface upper case letters, e.g., $\boldsymbol{H}$, $\boldsymbol{\Theta}$. The entry at $i$-th row, $j$-th column of a matrix $\boldsymbol{M}$ is denoted by $(\boldsymbol{M})_{ij}$ or $m_{ij}$. $\boldsymbol{I}_n$ or simply $\boldsymbol{I}$ denotes an $n \times n$ identity matrix; $\mathbf{1}_n$ and $\mathbf{0}_n$ denote $n$-tuple vectors of ones and zeros, respectively. We denote the trace of a square matrix $\boldsymbol{K}$ by $\mathrm{Tr}[\boldsymbol{K}]$. The pseudo-inverse of a matrix $\boldsymbol{U}$ is denoted by $\boldsymbol{U}^\dagger$. We denote finite or infinite sets by calligraphy letters, e.g., $\mathcal{H}$, $\mathcal{A}$.

Figure 2: Our IRepL model consists of three components: i) An $r$-dimensional encoder $\boldsymbol{f}$ belonging to the universal RKHS $\mathcal{H}_X$. ii) A measure of dependence that accounts for all dependence modes between data representation $Z$ and semantic attribute $S$ induced by the covariance between $Z = \boldsymbol{f}(X)$ and $\beta_S(S)$ where $\beta_S$ belongs to a universal RKHS $\mathcal{H}_S$. iii) A measure of dependency between $Z$ and the target attribute $Y$ defined similarly as that for $S$.

## 3.2 Problem Setup

Consider the probability space $(\Omega, \mathcal{F}, \mathbb{P})$, where $\Omega$ is the sample space, $\mathcal{F}$ is a $\sigma-$algebra on $\Omega$, and $\mathbb{P}$ is a probability measure on $\mathcal{F}$. We assume that the joint RV, $(X, Y, S)$ containing the input data $X \in \mathbb{R}^{d_X}$, the target label $Y \in \mathbb{R}^{d_Y}$, and the semantic attribute $S \in \mathbb{R}^{d_S}$, is a RV on $(\Omega, \mathcal{F})$ with joint distribution $\boldsymbol{p}_{X,Y,S}$. Furthermore, $Y$ and $S$ can also belong to any finite set, such as a categorical set. This setting enables us to work with both classification and multidimensional regression tasks, where the semantic attribute can be either categorical or multidimensional continuous/discrete RV.

**Assumption 1.** We assume that the encoder consists of $r$ functions from $\mathbb{R}^{d_X}$ to $\mathbb{R}$ in a universal RKHS $(\mathcal{H}_X, \, k_X(\cdot, \cdot))$ (e.g., RBF Gaussian kernel), where universality ensures that $\mathcal{H}_X$ can approximate any Borel function with arbitrary precision (Sriperumbudur et al., 2011).

Hence, the representation RV $Z$ can be expressed as

$$Z = \boldsymbol{f}(X) := [Z_1, \cdots, Z_r]^T \in \mathbb{R}^r, \quad Z_j = f_j(X), f_j \in \mathcal{H}_X \; \forall j = 1, \ldots, r, \tag{4}$$

where $r$ is the dimensionality of the representation. As we will discuss in Corollary 4.1, unlike common practice where $r$ is chosen on an ad-hoc basis, it is an object of interest for optimization. We consider a general scenario where both $Y$ and $S$ can be continuous/discrete or categorical, or one of $Y$ or $S$ is continuous/discrete while the other is categorical. To accomplish this, we replace the target loss, $\inf_{g_Y \in \mathcal{H}_Y} \mathbb{E}_{X,Y} [L_Y (g_Y(Z), Y)]$ in (2) by the negative of a non-parametric measure of dependence, i.e., $-\text{Dep}(Z, Y)$. The main reason for this replacement is that maximizing statistical dependency between the representation $Z$ and the target attribute $Y$ can flexibly learn a representation applicable to different downstream target tasks, including regression, classification, clustering, etc (Barshan et al., 2011). Particularly, Theorem 6 in Section 5.2 indicates that with an appropriate choice of involved RKHS for $\text{Dep}(Z, Y)$, we can learn a representation that lends itself to an estimator that performs as optimally as a Bayes estimator i.e., $\mathbb{E}_X[Y|X]$. Furthermore, in an unsupervised setting, where there is no target attribute $Y$, the target loss can be replaced by $\text{Dep}(Z, X)$, which implicitly forces the representation $Z$ to be as dependent on the input data $X$. This scenario is of practical interest when a data producer aims to provide an invariant representation for an unknown downstream target task.

## 4 Choice of Dependence Measure

We only discuss $\text{Dep}(Z, S)$ since we adopt the same dependence measure for $\text{Dep}(Z, Y)$. Accounting for all possible non-linear relations between RVs is a key desideratum of dependence measures. A well-known example of such measures is mutual information (MI) (e.g., MINE (Belghazi et al., 2018)). However, calculating MI for continuous multidimensional representations is analytically challenging and computationally intractable. Kernel-based measures are an alternative solution with the attractive properties of being computationally feasible/efficient and analytically tractable (Gretton et al., 2005b).

**Definition 2.** Let $\boldsymbol{D} = \{(\boldsymbol{x}_1, \boldsymbol{y}_1, \boldsymbol{s}_1), \cdots, (\boldsymbol{x}_n, \boldsymbol{y}_n, \boldsymbol{s}_n)\}$ be the training data, containing $n$ i.i.d. samples from the joint distribution $\boldsymbol{p}_{X,Y,S}$. Invoking the representer theorem (Shawe-Taylor & Cristianini, 2004), it follows that for

each $f_j \in \mathcal{H}_X$ $(j = 1, \cdots, r)$ we have $Z_j = f_j(X) = \sum_{i=1}^{n} \theta_{ji} k_X(\boldsymbol{x}_i, X)$ where $\theta_{ij}$s are the learnable linear weights. Consequently, it follows that

$$\boldsymbol{f}(X) = \boldsymbol{\Theta} \left[ k_X(\boldsymbol{x}_1, X), \cdots, k_X(\boldsymbol{x}_n, X) \right]^T, \tag{5}$$

where $\boldsymbol{\Theta} \in \mathbb{R}^{r \times n}$ and $(\boldsymbol{\Theta})_{ji} = \theta_{ji}$.

Principally, $Z \perp\!\!\!\perp S$ if and only if (iff) $\mathbb{C}\mathrm{ov}(\alpha(Z), \beta_S(S)) = 0$ for all Borel functions $\alpha : \mathbb{R}^r \to \mathbb{R}$ and $\beta_S : \mathbb{R}^{d_S} \to \mathbb{R}$ belonging to the universal RKHSs $\mathcal{H}_Z$ and $\mathcal{H}_S$, respectively. Alternatively, $Z \perp\!\!\!\perp S$ iff $\mathrm{HSIC}(Z, S) = 0$ where HSIC (Gretton et al., 2005a) is defined as

$$\mathrm{HSIC}(Z, S) := \sum_{\alpha \in \mathcal{U}_Z} \sum_{\beta_S \in \mathcal{U}_S} \mathbb{C}\mathrm{ov}^2 \left( \alpha(Z), \beta_S(S) \right), \tag{6}$$

where $\mathcal{U}_Z$ and $\mathcal{U}_S$ are countable orthonormal basis sets for the separable universal RKHSs $\mathcal{H}_Z$ and $\mathcal{H}_S$, respectively. However, since $Z = \boldsymbol{f}(X)$ where $\boldsymbol{f}$ is defined in (4), calculating $\mathbb{C}\mathrm{ov}(\alpha(Z), \beta_S(S))$ necessitates the application of a cascade of kernels, which limits the analytical tractability of our solution. Therefore, we adopt a simplified version of HSIC that considers transformation on $S$ only but affords analytical tractability for solving the IRepL optimization problem. We define this measure as

$$\mathrm{Dep}(Z, S) := \sum_{j=1}^{r} \sum_{\beta_S \in \mathcal{U}_S} \mathbb{C}\mathrm{ov}^2 \left( Z_j, \beta_S(S) \right), \tag{7}$$

where $Z_j = f_j(X)$ for $f_j$s defined in (4). We note that $\mathrm{Dep}(\cdot, \cdot)$, unlike HSIC and other kernelization-based dependence measures, is not symmetric. However, symmetry is not necessary for measuring statistical dependence. To guarantee the boundedness of $\mathrm{Dep}(Z, S)$ and $\boldsymbol{f}(X)$, we make the following assumption in the remainder of this paper.

**Assumption 2.** We assume that $(\mathcal{H}_S, k_S(\cdot, \cdot))$ and $(\mathcal{H}_Y, k_Y(\cdot, \cdot))$ are separable[2] and the kernel functions are bounded:

$$\mathbb{E}_U \left[ k_U(U, U) \right] < \infty, \quad \text{for } U = X, Y, S. \tag{8}$$

The measure $\mathrm{Dep}(Z, S)$ in (7) captures all modes of non-linear dependence under the assumption that the distribution of a low-dimensional projection of high-dimensional data is approximately normal (Diaconis & Freedman, 1984), (Hall & Li, 1993). To see why this reasoning is relevant, we note from (5) that $Z$ can be expressed as $Z = \boldsymbol{\Theta}V$, where $V \in \mathbb{R}^n$ and $\boldsymbol{\Theta} \in \mathbb{R}^{r \times n}$. This indicates that for large $n$ and small $r$ (which is the case for most real-world datasets), $Z$ is indeed a low-dimensional projection of high-dimensional data. In other words, $(Z, \beta_S(S))$ is approximately a jointly Gaussian RV. In our numerical experiments in Section 6, we empirically observe that $\mathrm{Dep}(Z, S)$ enjoys an almost monotonic relation with the underlying invariance measure and captures all modes of dependency in practice, especially as $Z \perp\!\!\!\perp S$. Nevertheless, if the normality assumption on the distribution of $(Z, \beta_S(S))$ fails, $\mathrm{Dep}(Z, S)$ reduces to measuring the linear dependency between $Z$ and $\beta_S(S)$ for all Borel functions $\beta_S$. This corresponds to measuring the mean independency of $Z$ from $S$, i.e., how much information a predictor (linear and non-linear) can infer (in the sense of MSE) about $Z$ from $S$. See Appendix H for more technical details on mean independency.

**Lemma 1.** Let $\boldsymbol{K}_X, \boldsymbol{K}_S \in \mathbb{R}^{n \times n}$ be the Gram matrices corresponding to $\mathcal{H}_X$ and $\mathcal{H}_S$, respectively, i.e., $(\boldsymbol{K}_X)_{ij} = k_X(\boldsymbol{x}_i, \boldsymbol{x}_j)$ and $(\boldsymbol{K}_S)_{ij} = k_S(\boldsymbol{s}_i, \boldsymbol{s}_j)$, where covariance is empirically estimated as

$$\mathbb{C}\mathrm{ov} \left( f_j(X), \beta_S(S) \right) \approx \frac{1}{n} \sum_{i=1}^{n} f_j(\boldsymbol{x}_i) \beta_S(\boldsymbol{s}_i) - \frac{1}{n^2} \sum_{i=1}^{n} \sum_{k=1}^{n} f_j(\boldsymbol{x}_i) \beta_S(\boldsymbol{s}_k).$$

It follows that, the corresponding empirical estimator for $\mathrm{Dep}(Z, S)$ is

$$\mathrm{Dep}^{\mathrm{emp}}(Z, S) = \frac{1}{n^2} \left\| \boldsymbol{\Theta} \boldsymbol{K}_X \boldsymbol{H} \boldsymbol{L}_S \right\|_F^2, \tag{9}$$

where $\boldsymbol{H} = \boldsymbol{I}_n - \frac{1}{n} \boldsymbol{1}_n \boldsymbol{1}_n^T$ is the centering matrix, $\boldsymbol{L}_S$ is a full column-rank matrix in which $\boldsymbol{L}_S \boldsymbol{L}_S^T = \boldsymbol{K}_S$ (Cholesky factorization), and $\boldsymbol{K}_S$ is the Gram matrix corresponding to $\mathcal{H}_S$. Furthermore, the empirical estimator in (9) has a bias of $\mathcal{O}(n^{-1})$ and a convergence rate of $\mathcal{O}(n^{-1/2})$.

---

[2]A Hilbert space is separable iff it has a countable orthonormal basis set.

*Proof.* The main idea for proving equality (9) is to employ the represener theorem to express $f_j$ and $\beta_S$. For the bias and convergence rate, one can express equation (9) in terms of trace and then employ U-statistics together with Hoeffdig's inequality (Hoeffding, 1994). For complete proof, see Appendix B. $\qquad\square$

Finally, we note that the dependence measure between $Z$ and $Y$ can be defined similarly.

## 5  Exact Kernelized Trade-Off

Consider the optimization problem corresponding to $\mathcal{T}_{\text{Opt}}$ in (2). Recall that $Z = \boldsymbol{f}(X)$ is an $r$-dimensional RV, where the embedding dimensionality $r$ is also a variable to be optimized. A common desideratum of learned representations is that of compactness (Bengio et al., 2013) to avoid learning representations with redundant information where different dimensions are highly correlated to each other. Therefore, going beyond the assumption that each component of $\boldsymbol{f}$ (i.e., $f_j$s) belongs to the universal RKHS $\mathcal{H}_X$, we impose additional constraints on the representation. Specifically, we constrain the search space of the encoder $\boldsymbol{f}(\cdot)$ to learn a disentangled representation (Bengio et al., 2013) as follows

$$\mathcal{A}_r := \left\{ (f_1, \cdots, f_r) \,\middle|\, f_i, f_j \in \mathcal{H}_X, \, \mathbb{Cov}\left(f_i(X), f_j(X)\right) + \gamma \left\langle f_i, f_j \right\rangle_{\mathcal{H}_X} = \delta_{i,j} \right\}. \tag{10}$$

In the above set, the $\mathbb{Cov}\left(f_i(X), f_j(X)\right)$ part enforces the covariance matrix of $Z = \boldsymbol{f}(X)$ to be an identity matrix. Such disentanglement is also used in principal component analysis (PCA). It encourages the variance of each entry of $Z$ to be one and different entries of $Z$ to be uncorrelated with each other. The regularization part, $\gamma \left\langle f_i, f_j \right\rangle_{\mathcal{H}_X}$ encourages the encoder components to be as orthogonal as possible to each other and to be of unit norm, which aids with numerical stability during empirical estimation (Fukumizu et al., 2007). As the following theorem states formally, such disentanglement is an invertible transformation; therefore, it does not nullify any information.

**Theorem 2.** Let $Z = \boldsymbol{f}(X)$ be an arbitrary representation of the input data, where $\boldsymbol{f} \in \mathcal{H}_X$. Then, there exists an invertible Borel function $\boldsymbol{h}$, such that $\boldsymbol{h} \circ \boldsymbol{f}$ belongs to $\mathcal{A}_r$.

*Proof.* The main idea is to search for an explicit expression for $\boldsymbol{h}$ in terms of the invertible operator $\Sigma_{XX} + \gamma I_X$, where $\Sigma_{XX}$ is induced by the bi-linear functional $\mathbb{Cov}\left(f_i(X), f_j(X)\right) = \left\langle \Sigma_{XX} f_i, f_j \right\rangle_{\mathcal{H}_X}$, and $I_X$ is the identity operator from $\mathcal{H}_X$ to itself. See Appendix C for complete proof. $\qquad\square$

This Theorem implies that the disentanglement preserves the performance of the downstream task since any target network can revert the disentanglement $\boldsymbol{h}$ and access the original representation $Z$. In addition, any deterministic measurable transformation of $Z$ will not add any information about $S$ that does not already exist in $Z$.

We define our $\text{K}-\mathcal{T}_{\text{Opt}}$ as

$$\sup_{\boldsymbol{f} \in \mathcal{A}_r} \left\{ J\left(\boldsymbol{f}, \lambda\right) := (1 - \lambda) \operatorname{Dep}\left(\boldsymbol{f}(X), Y\right) - \lambda \operatorname{Dep}\left(\boldsymbol{f}(X), S\right) \right\}, \quad 0 \le \lambda < 1, \tag{11}$$

where $\lambda$ is the utility-invariance trade-off parameter. Fortunately, the above optimization problem lends itself to a closed-form solution.

**Theorem 3.** Consider the operator $\Sigma_{SX}$ to be induced by the bi-linear functional $\mathbb{Cov}(\alpha(X), \beta_S(S)) = \left\langle \Sigma_{SX}\alpha, \beta_S \right\rangle_{\mathcal{H}_S}$ and define $\Sigma_{YX}$ and $\Sigma_{XX}$, similarly. Then, a global optimizer for the optimization problem in (11) is the eigenfunctions corresponding to the $r$ largest eigenvalues of the following generalized eigenvalue problem

$$\left((1 - \lambda) \Sigma_{YX}^* \Sigma_{YX} - \lambda \Sigma_{SX}^* \Sigma_{SX}\right) \boldsymbol{f} = \tau \left(\Sigma_{XX} + \gamma I_X\right) \boldsymbol{f}, \tag{12}$$

where $\gamma$ is the disentanglement regularization parameter defined in (10), and $\Sigma^*$ is the adjoint of $\Sigma$.

*Proof.* The first step is to express $\operatorname{Dep}(f(X), Y)$ and $\operatorname{Dep}(f(X), S)$ in terms of $\Sigma_{YX}$ and $\Sigma_{SX}$, respectively. The resulting expression can be restated as a generalized Rayleigh quotient (Strawderman, 1999) which can be solved via a generalized eigenvalue formulation. For complete proof, see Appendix D. $\qquad\square$

**Remark 1.** If the trade-off parameter $\lambda = 0$ (i.e., no semantic independence constraint is imposed) and $\gamma \to 0$, the solution in Theorem 3 is equivalent to a supervised kernel-PCA. On the other hand, if $\lambda \to 1$ (i.e., utility is ignored and only semantic independence is considered), the solution in Theorem 3 is the eigenfunctions corresponding to the $r$ smallest eigenvalues of $\Sigma_{SX}^* \Sigma_{SX}$, which are the directions that are the least explanatory of the semantic attribute $S$.

Now, consider the empirical counterpart of the optimization problem (11),

$$\sup_{\boldsymbol{f} \in \mathcal{A}_r} \left\{ J^{\text{emp}}(\boldsymbol{f}, \lambda) := (1 - \lambda) \operatorname{Dep}^{\text{emp}}(\boldsymbol{f}(X), Y) - \lambda \operatorname{Dep}^{\text{emp}}(\boldsymbol{f}(X), S) \right\}, \quad 0 \le \lambda < 1 \tag{13}$$

where $\operatorname{Dep}^{\text{emp}}(\boldsymbol{f}(X), S)$ is given in (9) and $\operatorname{Dep}^{\text{emp}}(\boldsymbol{f}(X), Y)$ is defined similarly.

**Theorem 4.** Let the Cholesky factorization of $\boldsymbol{K}_X$ be $\boldsymbol{K}_X = \boldsymbol{L}_X \boldsymbol{L}_X^T$, where $\boldsymbol{L}_X \in \mathbb{R}^{n \times d}$ ($d \le n$) is a full column-rank matrix. Let $r \le d$, then a solution to (13) is

$$\boldsymbol{f}^{\text{opt}}(X) = \boldsymbol{\Theta}^{\text{opt}} \left[ k_X(\boldsymbol{x}_1, X), \cdots, k_X(\boldsymbol{x}_n, X) \right]^T$$

where $\boldsymbol{\Theta}^{\text{opt}} = \boldsymbol{U}^T \boldsymbol{L}_X^\dagger$ and the columns of $\boldsymbol{U}$ are eigenvectors corresponding to the $r$ largest eigenvalues of the following generalized eigenvalue problem.

$$\boldsymbol{L}_X^T \left( (1 - \lambda) \boldsymbol{H} \boldsymbol{K}_Y \boldsymbol{H} - \lambda \boldsymbol{H} \boldsymbol{K}_S \boldsymbol{H} \right) \boldsymbol{L}_X \boldsymbol{u} = \tau \left( \frac{1}{n} \boldsymbol{L}_X^T \boldsymbol{H} \boldsymbol{L}_X + \gamma \boldsymbol{I} \right) \boldsymbol{u}. \tag{14}$$

Further, the objective value of (13) is equal to $\sum_{j=1}^r \tau_j$, where $\{\tau_1, \cdots, \tau_r\}$ are the $r$ largest eigenvalues of (14).

*Proof.* The main idea is to employ empirical expressions obtained in equality 9. The resulting expression on (13) can be expressed as a trace optimization problem which can be solved by eigenvalue formulation (Kokiopoulou et al., 2011). See Appendix E for detailed proof. □

**Corollary 4.1.** *Embedding Dimensionality*: A useful corollary of Theorem 4 is characterizing optimal embedding dimensionality as a function of the trade-off parameter, $\lambda$:

$$r^{\text{Opt}}(\lambda) := \arg \sup_{0 \le r \le d} \left\{ \sup_{\boldsymbol{f} \in \mathcal{A}_r} \left\{ J^{\text{emp}}(\boldsymbol{f}, \lambda) \right\} \right\} = \text{ number of non-negative eigenvalues of (14).}$$

To examine these results, consider two extreme cases: i) If there is no semantic independence constraint (i.e., $\lambda = 0$), all eigenvalues of (14) are non-negative since $\boldsymbol{H} \boldsymbol{K}_Y \boldsymbol{H}$ is a non-negative definite matrix and $\frac{1}{n} \boldsymbol{L}_X^T \boldsymbol{H} \boldsymbol{L}_X + \gamma \boldsymbol{I}$ is a positive definite matrix. This indicates that $r^{\text{Opt}}$ is equal to the maximum possible value (that is equal to $d$), and therefore it is not required for $Z$ to nullify any information in $X$. ii) If we are only concerned about semantic independence and want to ignore the target task utility (i.e., $\lambda \to 1$), all eigenvalues of (14) are non-positive and therefore $r^{\text{Opt}}$ would be the number of zero eigenvalues of (14). This indicates that $\operatorname{Dep}^{\text{emp}}(Z, S)$ in (9) is equal to zero, since $\boldsymbol{\Theta}^{\text{opt}} \boldsymbol{K}_X$ is zero for zero eigenvalues of (14) when $\lambda \to 1$. In this case, adding more dimension to $Z$ will necessarily increase $\operatorname{Dep}^{\text{emp}}(Z, S)$.

The following Theorem characterizes the convergence behavior of empirical $\text{K} - \mathcal{T}_{\text{Opt}}$ to its population counterpart.

**Theorem 5.** Assume that $k_S$ and $k_Y$ are bounded by one and $f_j^2(\boldsymbol{x}_i) \le M$ for any $j = 1, \ldots, r$ and $i = 1, \ldots, n$ for which $\boldsymbol{f} = (f_1, \ldots, f_r) \in \mathcal{A}_r$. Then, for any $n > 1$ and $0 < \delta < 1$, with probability at least $1 - \delta$, we have

$$\left| \sup_{\boldsymbol{f} \in \mathcal{A}_r} J(\boldsymbol{f}, \lambda) - \sup_{\boldsymbol{f} \in \mathcal{A}_r} J^{\text{emp}}(\boldsymbol{f}, \lambda) \right| \le rM \sqrt{\frac{\log(6/\delta)}{0.22^2 \, n}} + \mathcal{O}\left(\frac{1}{n}\right).$$

*Proof.* This Theorem can be proved by using the results of the convergence rate of $\operatorname{Dep}(Z, S)$ in Lemma 1. For detailed proof, see Appendix F. □

Note that, for any $\boldsymbol{x}$ in the training set, $f_j(\boldsymbol{x})$ can be calculated as $f_j(\boldsymbol{x}) = \sum_{i=1}^n \theta_{ji} k_X(\boldsymbol{x}_i, \boldsymbol{x})$. We can assume that $k_X(\cdot, \cdot)$ is bounded. For example, in RBF Gaussian and Laplacian RKHSs, both of which are universal, $k_X(\cdot, \cdot) \le 1$. This implies that $f_j^2(\boldsymbol{x}) \le \sqrt{n} \|\boldsymbol{\theta}_j\|$, where $\boldsymbol{\theta}_j$ is $j$-th row of $\boldsymbol{\Theta}$ in equation (5). One always can normalize $f_j(\boldsymbol{x})$ by dividing it by the maximum of $\sqrt{n} \|\boldsymbol{\theta}_j\|$ over $j$s, or by dividing by the maximum of $|f_j(\boldsymbol{x}_i)|$ over $i$s and $j$s. Notice that this normalization is only a scalar multiplication and has no effect on the invariance of $Z = \boldsymbol{f}(X)$ to $S$ and the utility of any downstream target task predictor $g_Y(Z)$.

### 5.1 Numerical Complexity

**Computational Complexity:** If $\boldsymbol{L}_X$ in (14) is provided in the training dataset, then the computational complexity of obtaining the optimal encoder is $\mathcal{O}(l^3)$, where $l \leq n$ is the numerical rank of the Gram matrix $\boldsymbol{K}_X$. However, the dominating part of the computational complexity is due to the Cholesky factorization, $\boldsymbol{K}_X = \boldsymbol{L}_X \boldsymbol{L}_X^T$, which is $\mathcal{O}(n^3)$. Using random Fourier features (RFF) (Rahimi et al., 2007), $k_X(\boldsymbol{x}, \boldsymbol{x}')$ can be approximated by $\boldsymbol{r}_X(\boldsymbol{x})^T \boldsymbol{r}_X(\boldsymbol{x}')$, where $\boldsymbol{r}_X(\boldsymbol{x}) \in \mathbb{R}^d$. In this situation, the Cholesky factorization can be directly calculated as

$$\boldsymbol{L}_X = \begin{bmatrix} \boldsymbol{r}_X(\boldsymbol{x}_1)^T \\ \vdots \\ \boldsymbol{r}_X(\boldsymbol{x}_n)^T \end{bmatrix} \in \mathbb{R}^{n \times d}. \tag{15}$$

As a result, the computational complexity of obtaining the optimal encoder becomes $\mathcal{O}(d^3)$, where the RFF dimension, $d$, can be significantly less than the sample size $n$ with negligible error on the approximation $k_X(\boldsymbol{x}, \boldsymbol{x}') \approx \boldsymbol{r}_X(\boldsymbol{x})^T \boldsymbol{r}_X(\boldsymbol{x}')$.

**Memory Complexity:** The memory complexity of (14), if calculated naively, is $\mathcal{O}(n^2)$ since $\boldsymbol{K}_Y$ and $\boldsymbol{K}_S$ are $n$ by $n$ matrices. However, using RFF together with Cholesky factorization $\boldsymbol{K}_Y = \boldsymbol{L}_Y \boldsymbol{L}_Y^T$, $\boldsymbol{K}_S = \boldsymbol{L}_S \boldsymbol{L}_S^T$, the left-hand side of (14) can be re-arranged as

$$(1 - \lambda) \left( \boldsymbol{L}_X^T \tilde{\boldsymbol{L}}_Y \right) \left( \tilde{\boldsymbol{L}}_Y^T \boldsymbol{L}_X \right) - \lambda \left( \boldsymbol{L}_X^T \tilde{\boldsymbol{L}}_S \right) \left( \tilde{\boldsymbol{L}}_S^T \boldsymbol{L}_X \right), \tag{16}$$

where $\tilde{\boldsymbol{L}}_Y^T = \boldsymbol{H} \boldsymbol{L}_Y = \boldsymbol{L}_Y - \frac{1}{n} \mathbf{1}_n (\mathbf{1}_n^T \boldsymbol{L}_Y)$ and therefore, the required memory complexity is $\mathcal{O}(nd)$. Note that $\tilde{\boldsymbol{L}}_S^T$ and $\boldsymbol{H} \boldsymbol{L}_X$ can be calculated similarly.

### 5.2 Target Task Performance in K$-\mathcal{T}_{\text{Opt}}$

Assume that the desired target loss function is MSE. Then, in the following Theorem, we show that maximizing $\text{Dep}(\boldsymbol{f}(X), Y)$ over $\boldsymbol{f} \in \mathcal{A}_r$ can learn a representation $Z$ that is informative enough for a target predictor on $Z$ to achieve the most optimal estimation, i.e., the Bayes estimator ($\mathbb{E}[Y \mid X]$).

**Theorem 6.** Let $\boldsymbol{f}^*$ be the optimal encoder by maximizing $\text{Dep}(\boldsymbol{f}(X), Y)$, where $\gamma \to 0$ and $\mathcal{H}_Y$ is a linear RKHS. Then, there exist $\boldsymbol{W} \in \mathbb{R}^{d_Y \times r}$ and $\boldsymbol{b} \in \mathbb{R}^{d_Y}$ such that $\boldsymbol{W} \boldsymbol{f}^*(X) + \boldsymbol{b}$ is the Bayes estimator, i.e.,

$$\begin{aligned} \mathbb{E}_{X,Y} \left[ \|\boldsymbol{W} \boldsymbol{f}(X)^* + \boldsymbol{b} - Y\|^2 \right] &= \inf_{h \text{ is Borel}} \mathbb{E}_{X,Y} \left[ \|h(X) - Y\|^2 \right] \\ &= \mathbb{E}_{X,Y} \left[ \|\mathbb{E}[Y \mid X] - Y\|^2 \right]. \end{aligned}$$

*Proof.* This Theorem corresponds to a linear least square target predictor on top of a kernelized encoder with a solution(s) similar to supervised kernel-PCA. Recall from Remark 1 that if $\lambda = 0$, optimizing $\text{Dep}(f(X), Y)$ will result in a supervised kernel-PCA. See Appendix G for formal proof. □

This Theorem implies that not only can $\text{Dep}(\boldsymbol{f}(X), Y)$ preserve all the necessary information in $Z$ to predict $Y$ optimally, the learned representation is simple enough for a linear regressor to achieve optimal performance.

## 6 Experiments

In this section, we numerically quantify our K$-\mathcal{T}_{\text{Opt}}$ through the closed-form solution for the encoder obtained in Section 5 on an illustrative toy example and two real-world datasets, Folktables and CelebA.

### 6.1 Baselines

We consider two types of baselines: (1) ARL (the main framework for IRepL) with MSE or Cross-Entropy as the adversarial loss. Such methods are expected to fail to learn a fully invariant representation (Adeli et al., 2021; Grari et al., 2020). These include (Xie et al., 2017; Zhang et al., 2018; Madras et al., 2018), and SARL (Sadeghi et al.,

2019). Among these baselines, except for SARL, all baselines are optimized via iterative minimax optimization, which is often unstable and not guaranteed to converge. On the other hand, SARL obtains a closed-form solution for the global optima of the minimax optimization under a linear dependence measure between $Z$ and $S$, which may fail to capture all modes of dependence between $Z$ and $S$. (2) HSIC-based adversarial loss that accounts for all modes of dependence, and as such, is theoretically expected to learn a fully invariant representation (Quadrianto et al., 2019). However, since stochastic gradient descent is used for learning, it lacks convergence guarantees to the global optima.

## 6.2  Datasets

**Gaussian Toy Example:** We design an illustrative toy example where $X$ and $S$ are mean independent in some dimensions but not fully independent in those dimensions. Specifically, $X$ and $S$ are 4-dimensional continuous RVs and generated as follows,

$$U = [U_1, U_2, U_3, U_4] \sim \mathcal{N}(\mathbf{0}_4, \boldsymbol{I}_4), \quad N \sim \mathcal{N}(\mathbf{0}_4, \boldsymbol{I}_4), \quad U \perp\!\!\!\perp N$$
$$X = \cos\left(\frac{\pi}{6} U\right) + 0.005N, \;\; S = \left[\sin\left(\frac{\pi}{6}[U_1, U_2]\right), \cos\left(\frac{\pi}{6}[U_3, U_4]\right)\right], \tag{17}$$

where $\sin(\cdot)$ and $\cos(\cdot)$ are applied point-wise. To generate the target attribute, we define four binary RVs as follows.

$$Y_i = \mathbf{1}_{\{|U_i|>T\}}(U_i), \;\; i = 1, 2, 3, 4,$$

where $\mathbf{1}_{\mathcal{B}}(\cdot)$ is the indicator function, and we set $T = 0.6744$, so it holds that $\mathbb{P}[Y_i = 0] = \mathbb{P}[Y_i = 1] = 0.5$ for $i = 1, 2, 3, 4$. Finally, we define $Y$ as a 16-class categorical RV concatenated by $Y_i$s. Since $S$ is dependent on $X$ through all the dimensions of $X$, a wholly invariant $Z$ (i.e., $Z \perp\!\!\!\perp S$) should not contain any information about $X$. However, since $[S_1, S_2]$ is only mean independent of $[X_1, X_2]$ (i.e., $\mathbb{E}\left[S_1, S_2 \,\middle|\, X_1, X_2\right] = \mathbb{E}\left[S_1, S_2\right]$), ARL baselines with MSE as the adversary loss, i.e., Xie et al. (2017); Zhang et al. (2018); Madras et al. (2018) and SARL cannot capture the dependency of $Z$ to $[S_1, S_2]$ and result in a representation that is always dependent on $[S_1, S_2]$ (see Section H for theoretical details). We sample $18,000$ instances from $\boldsymbol{p}_{X,Y,S}$ independently and split these samples equally into training, validation, and testing partitions.

**Folktables:** We consider a fair representation learning task on Folktables (Ding et al., 2021) dataset (a derivation of the US census data). Specifically, we consider 2018-WA (Washington) and 2018-NY (New York) census data where the target attribute $Y$ is the employment status (binary for WA and 4 categories for NY and the semantic attribute $S$ is age (discrete value between 0 and 95 years). We seek to learn a representation that predicts employment status while being fair in demographic parity (DP) w.r.t. age. DP requires that the prediction $\widehat{Y}$ be independent of $S$, which can be achieved by enforcing $Z \perp\!\!\!\perp S$. The WA and NY datasets contain $76,225$ and $196,967$ samples, each constructed from 16 different features. We randomly split the data into training ($70\%$), validation ($15\%$), and testing ($15\%$) partitions. Further, we adopt embeddings for categorical features (learned in a supervised fashion to predict $Y$) and normalization for continuous/discrete features (dividing by the maximum value).

**CelebA:** CelebA dataset (Liu et al., 2015) contains $202,599$ face images of $10,177$ different celebrities with standard training, validation, and testing splits. Each image is annotated with 40 different attributes. We choose the target attribute $Y$ as the high cheekbone attribute (binary) and the semantic attribute $S$ as the concatenation of gender and age (a 4-class categorical RV). The objective of this experiment is similar to that of Folktables. Since raw image data is not appropriate for kernel methods, we pre-train a ResNet-18 (He et al., 2016) (supervised to predict $Y$) on CelebA images and extract features of dimension 256. These features are used as the input data for all methods.

## 6.3  Evaluation Metrics

We use the accuracy of the classification tasks (16-class classification for Gaussian toy example, employment prediction for Folktables, and high cheekbone prediction for CelebA) as the utility metric. For Folktables and CelebA datasets, we define DP violation as

$$\text{DPV}(\widehat{Y}, S) := \mathbb{E}_{\widehat{Y}}\left[\mathbb{V}\text{ar}_S\left(\mathbb{P}[\widehat{Y}\,|\,S]\right)\right] \tag{18}$$

and use it as a metric to measure the variance (unfairness) of the prediction $\widehat{Y}$ w.r.t. the semantic attribute $S$. For the Gaussian toy example, the above metric is not suitable since $S$ is a continuous RV. To circumvent this difficulty, we

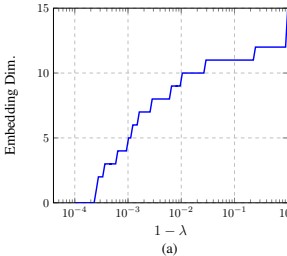 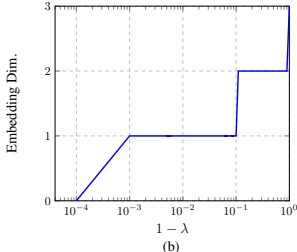

Figure 3: Plots of $r^{\mathrm{Opt}}(\lambda)$ versus the dependence trade-off parameter $1 - \lambda$ for (a) the Gaussian toy dataset and (b) Folktables-NY dataset. There is a non-decreasing relation between $r^{\mathrm{Opt}}(\lambda)$ and $1 - \lambda$.

employ KCC (Bach & Jordan, 2002)

$$\mathrm{KCC}(Z, S) := \sup_{\alpha \in \mathcal{H}_Z, \beta \in \mathcal{H}_S} \frac{\mathbb{C}\mathrm{ov}(\alpha(Z), \beta(S))}{\sqrt{\mathbb{V}\mathrm{ar}(\alpha(Z))\,\mathbb{V}\mathrm{ar}(\beta(S))}}, \tag{19}$$

as a measure of invariance of $Z$ to $S$, where $\mathcal{H}_Z$ and $\mathcal{H}_S$ are RBF-Gaussian RKHS. The reason for using KCC instead of HSIC is that, unlike HSIC, KCC is normalized. Therefore it is a more readily interpretable measure for comparing the invariance of representations between different methods.

### 6.4 Choice of $(Y, S)$ Pair

The existence of a utility-invariance trade-off ultimately depends on the statistical dependency between target and semantic attributes. If $\mathrm{Dep}(Z, S)$ is negligible, a trade-off does not exist. Keeping this in mind, we first chose the semantic attribute to be a sensitive attribute for Folktables (i.e., age) and CelebA (i.e., concatenation of age and gender) datasets. Then, we calculated the data imbalance (i.e., $|\mathbb{P}[Y = 0] - 0.5|$) and $\mathrm{KCC}(Y, S)$ for all possible $Y$s. Finally, we chose $Y$ with a small data imbalance and a moderate $\mathrm{KCC}(Y, S)$. For Folktables dataset, $|\mathbb{P}[\mathrm{employment} = 0] - 0.5| = 0.04$ and $\mathrm{KCC}(\mathrm{employment}, \mathrm{age}) = 0.4$. For CelebA dataset, $|\mathbb{P}[\mathrm{high\ cheekbone} = 0] - 0.5| = 0.05$ and $\mathrm{KCC}(\mathrm{high\ cheekbone}, [\mathrm{age}, \mathrm{gender}]) = 0.1$.

### 6.5 Implementation Details

For all methods, we pick different values of $\lambda$ (100 $\lambda$s for the Gaussian toy example and 70 $\lambda$s for Folktables and CelebA datasets) between zero and one for obtaining the utility-invariance trade-off. We train the baselines that use a neural network for encoder five times with different random seeds. We let the random seed also change the training-validation-testing split for the Folktables dataset (CelebA and Gaussian datasets have fixed splits).

**Embedding Dimensionality**: None of the baseline methods have any strategy to find the optimum embedding dimensionality ($r$), and they all set $r$ to a constant w.r.t. $\lambda$. Therefore, for baseline methods, we set $r = 15$ (i.e., the minimum dimensionality required to classify 16 different categories linearly) for the Gaussian toy example and $r = 3$ (i.e., the minimum dimensionality required to classify 4 different categories linearly) for Folktables-NY dataset, that is also equal to $r^{\mathrm{Opt}}$ when $\lambda = 0$. For K-$\mathcal{T}_{\mathrm{Opt}}$, we use $r^{\mathrm{Opt}}(\lambda)$ in Corollary 4.1. See Figure 3 for the plot of $r^{\mathrm{Opt}}$ versus $\lambda$ for the toy Gaussian and Folktables-NY datasets. For Folktables-WA and CelebA datasets, $r^{\mathrm{Opt}}(\lambda = 0)$ is equal to one, and therefore we let $r = 1$ for all methods and all $0 \leq \lambda < 1$.

**K$-\mathcal{T}_{\mathrm{Opt}}$ (Ours)**: We let $\mathcal{H}_X$, $\mathcal{H}_S$, and $\mathcal{H}_Y$ be RBF Gaussian RKHS, where we compute the corresponding band-widths (i.e., $\sigma$s) using the median strategy introduced by Gretton et al. (2007). We optimize the regularization parameter $\gamma$ in the disentanglement set (10) by minimizing the corresponding target losses over $\gamma$s in $\{10^{-6}, 10^{-5}, 10^{-4}, 10^{-3}, 10^{-2}, 10^{-1}, 1\}$ on validation sets. RFF (as discussed in Section 5.1) is adopted for all datasets. For RFF dimensionality, we started with a small value. Then, we gradually increased it until we reached the maximum possible performance for $\lambda = 0$ (i.e., the standard unconstrained representation learning) on the corresponding validation sets. Through this process, the final RFF dimensionality is 100 for the Gaussian dataset, 5000 for the Folktables dataset, and 1000 for the CelebA dataset.

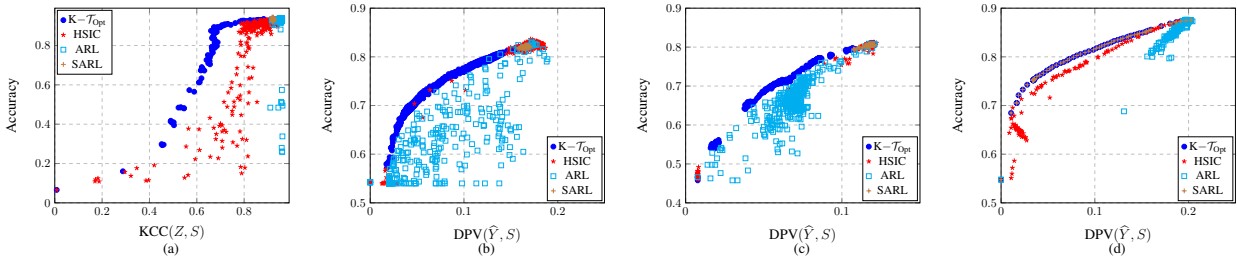

Figure 4: Utility versus invariance trade-offs obtained by K$-\mathcal{T}_{\text{Opt}}$ and other baselines for (a) Gaussian, (b) Folktables-WA, (c) Folktables-NY, and (d) CelebA datasets. K-$\mathcal{T}_{\text{Opt}}$ stably spans the entire trade-off front and considerably dominates other methods for all datasets. (a) ARL and SARL Sadeghi et al. (2019) span a small portion of the trade-off front since $S$ is mean independent (but not fully independent) of $X$ in some dimensions for the Gaussian toy example. Despite using a universal dependence measure, HSIC-IRepL Quadrianto et al. (2019) performs sub-optimally due to the lack of convergence guarantees to the global optima.

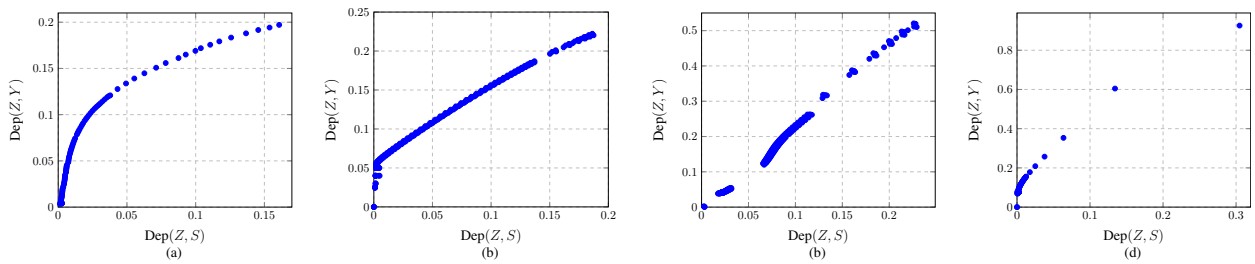

Figure 5: Dep$(Z, Y)$ versus Dep$(Z, S)$ in K$-\mathcal{T}_{\text{Opt}}$ for (a) Gaussian, (b) Folktables-WA, (c) Folktables-NY, and (d) CelebA datasets. We can observe that there is the same trend in Dep$(Z, Y)$-Dep$(Z, S)$ trade-off as utility-invariance-trade-off in Figure 4.

**SARL (Sadeghi et al., 2019)**: SARL method is similar to our K$-\mathcal{T}_{\text{Opt}}$ except that $\mathcal{H}_Y$ and $\mathcal{H}_S$ are linear RKHSs, and therefore we set $\sigma_X$ and $\gamma$ similar to that of K$-\mathcal{T}_{\text{Opt}}$.

**ARL (Xie et al., 2017; Zhang et al., 2018; Madras et al., 2018)**: The representation $Z = \boldsymbol{f}(X)$ is extracted via the encoder $\boldsymbol{f}$, which is an MLP (4 hidden layers and 15, 15 neurons for Gaussian data; 3 hidden layers and 128, 64 neurons for Folktables and CelebA datasets). The architectural choices were based on starting with a single linear layer and gradually increasing the number of layers and neurons until over-fitting was observed. This results in the number of encoder parameters for the Gaussian toy example to be 735, while K$-\mathcal{T}_{\text{Opt}}$ has $100 = 100 * r^{\text{Opt}}(\lambda \to 1) \leq 100 * r^{\text{Opt}}(\lambda) \leq 100 * r^{\text{Opt}}(\lambda = 0) = 1500$. For Folktables and CelebA, number of parameters is $41,024$ and $15,616$, respectively, for ARL and 5000 and 1000 for K$-\mathcal{T}_{\text{Opt}}$. The representation $Z$ is then fed to a target task predictor $g_Y$ and a proxy adversary $g_S$, both of which are MLPs with 2 hidden layers with 16 neurons for Gaussian data and 2 hidden layers with 128 neurons for Folktables and CelebA. All involved networks ($\boldsymbol{f}, g_Y, g_S$) are trained end-to-end. We use stochastic gradient descent-ascent (SGDA) Xie et al. (2017) with AdamW (Loshchilov & Hutter, 2017) as an optimizer to alternately train the encoder, target predictor, and proxy adversary. We use a batch size of 500 for Gaussian data; and 128 for Folktables and CelebA. Then, the corresponding learning rates are optimized over $\left\{10^{-2}, 10^{-3}, 5 \times 10^{-4}, 10^{-4}, 10^{-5}\right\}$ by minimizing the target loss on the corresponding validation sets.

**HSIC-IRepL (Quadrianto et al., 2019)**: This method can be formulated as (2) where Dep$(Z, S)$ is replaced by HSIC$(Z, S)$. The encoder and target predictor networks have the same architecture as ARL. And we use stochastic gradient descent to train the involved neural networks.

## 6.6 Results

**Utility-Invariance Trade-offs**: Figures 4 and 5 show the utility-invariance and Dep$(Z, Y)$-Dep$(Z, S)$ trade-offs for the toy Gaussian, Folktables-WA, Folktables-NY, and CelebA datasets. The invariance measure for the Gaussian

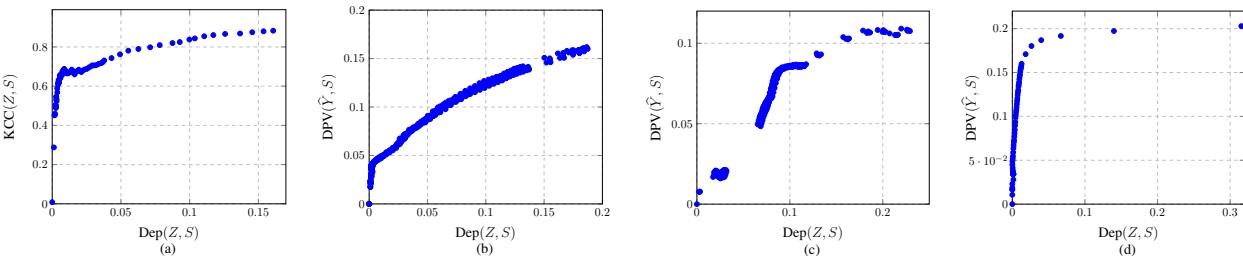

Figure 6: Invariance versus $\mathrm{Dep}(Z,S)$ of $\mathrm{K}-\mathcal{T}_{\mathrm{Opt}}$ for (a) Gaussian, (b) Folktables-WA, (c) Folktables-NY, and (d) CelebA datasets. $\mathrm{Dep}(Z,S)$ enjoys a monotonic relation with the underlying invariance measures.

toy example is KCC (19), and the invariance measure for Folktables and CelebA datasets is the fairness measure, DPV (18). We make the following observations: 1) $\mathrm{K}-\mathcal{T}_{\mathrm{Opt}}$ is highly stable and almost spans the entire trade-off front for all datasets except Folktables-NY, which can be due to the inability of scalarized single-objective formulation in (2), in contrast to the constrained optimization in (1), to find all Pareto-optimal points. 2) There is almost the same trend in the trade-off between $\mathrm{Dep}(Z,Y)$ and $\mathrm{Dep}(Z,S)$ (Figure 5) as the utility-invariance trade-off (Figure 4). This is a desirable observation since $\mathrm{Dep}(Z,Y)$-$\mathrm{Dep}(Z,S)$ trade-off is what we optimized in (11) as a surrogate to utility-invariance trade-off. 3) The baseline method HSIC-IRepL, despite using a universal dependence measure, leads to a sub-optimal trade-off front due to the lack of convergence guarantees to the global optima. 4) The baselines, ARL and SARL, span only a small portion of the trade-off front in the Gaussian toy example since some dimensions of the semantic attribute $S$ in (17) are mean independent (but not entirely independent) to some dimensions of $X$. Therefore the adversary does not provide any information to the encoder to discard $[S_1, S_2]$ from the representation. Moreover, in this dataset, ARL and SARL baselines do not approach $Z \perp\!\!\!\perp S$, i.e., $\mathrm{KCC}(Z,S) = 0$ cannot be attained for any value of the trade-off parameter $\lambda$. 5) ARL shows high deviation on the Folktables dataset due to the unstable nature of the minimax optimization. 6) SARL performs as well as our $\mathrm{K}-\mathcal{T}_{\mathrm{Opt}}$ for CelebA dataset. This is because both $S$ and $Y$ are categorical for the CelebA dataset, and therefore linear RKHS on one-hot encoded attribute performs just as well as universal RKHSs (Li et al., 2021).

**Universality of $\mathrm{Dep}(Z,S)$:** We empirically examine the practical validity of our assumption in Section 4 and verify if our dependence measure $\mathrm{Dep}(Z,S)$, defined in (7), can capture all modes of dependency between $Z$ and $S$. Figure 6 (a) shows the plot of the universal dependence measure $\mathrm{KCC}(Z,S)$ versus $\mathrm{Dep}(Z,S)$ for the Gaussian dataset and Figures 6 (b, c) illustrate the relationship between $\mathrm{DPV}(\widehat{Y},S)$ and $\mathrm{Dep}(Z,S)$ for Folktables and CelebA datasets, respectively. We observe a non-decreasing relation between the corresponding invariance measures and $\mathrm{Dep}(Z,S)$. More importantly, as $\mathrm{KCC}(Z,S) \to 0$ (or $\mathrm{DPV}(\widehat{Y},S) \to 0$) so does $\mathrm{Dep}(Z,S)$. These observations verify that $\mathrm{Dep}(Z,S)$ accounts for all modes of dependence between $Z$ and $S$.

## 6.7 Ablation Study

**Effect of Embedding Dimensionality:** In this experiment, we examine the significance of the embedding dimensionality, $r^{\mathrm{Opt}}(\lambda)$, discussed in Corollary 4.1. We obtain the utility-invariance trade-off when the embedding dimensionality is fixed to $r = r^{\mathrm{Opt}}(\lambda = 0) = 15$. A comparison between the utility-invariance trade-off induced by $r^{\mathrm{Opt}}(\lambda)$ and the fixed $r = 15$ is illustrated in Figure 7 (a). We observe that not only the utility-invariance trade-off for fixed $r$ is dominated by that of $r^{\mathrm{Opt}}(\lambda)$, but also, using fixed $r$ is unable to achieve the total invariance representation, i.e., $Z \perp\!\!\!\perp S$. Further, some of the largest eigenvalues of (14) versus the invariance trade-off parameter $\lambda$ are plotted in Figure 7 (b). We recall from Corollary 4.1 that, for any given $\lambda$, $r^{\mathrm{Opt}}$ is the number of non-negative eigenvalues of (14).

**Effect of Semantic Attribute Removal:** In this experiment, we examine the effect of removing $S$ (i.e., age) from the input data in the Folktables-WA dataset and examine whether this removal helps the utility-invariance trade-off. Figure 7 (c) shows the utility-invariance trade-off resulting from all methods, and Figure 7 (d) compares removing and keeping the age information from the input data for $\mathrm{K}-\mathcal{T}_{\mathrm{Opt}}$. Observe that: 1) There is almost the same trend in both keeping and removing the age attribute from the input data for all methods. 2) Removing the age attribute from input data slightly degrades the utility-invariance trade-off due to the lower information contained in the input data.

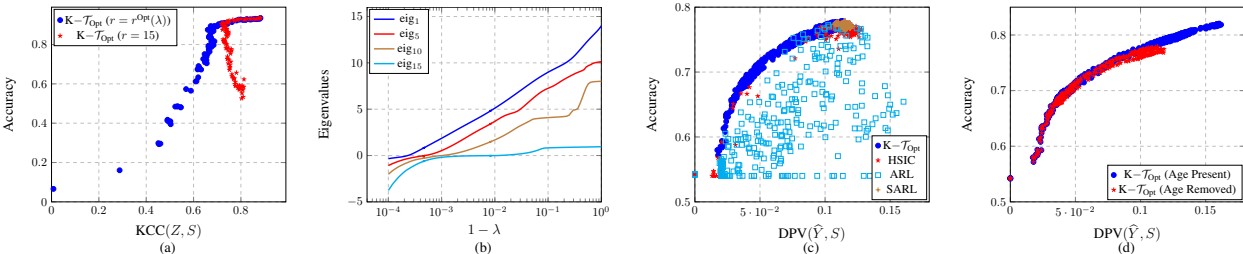

Figure 7: (a) Comparison between the utility-invariance trade-offs induced by the optimal embedding dimensionality $r^{\text{Opt}}(\lambda)$ and that of fixed $r = 15$. Fixed $r = 15$ is significantly dominated by that of $r^{\text{Opt}}(\lambda)$ and fails to attain $Z \perp\!\!\!\perp S$. (b) The first, fifth, tenth, and fifteenth largest eigenvalues in (14) versus $1 - \lambda$. Given $\lambda$, $r^{\text{Opt}}$ is equal to the number of non-negative eigenvalues. As $1 - \lambda$ decreases, the largest eigenvalues approach negative numbers. (c) Utility versus invariance trade-offs for all methods when age (i.e., the sensitive attribute) is discarded from the input data. (d) A comparison between trade-offs of K$-\mathcal{T}_{\text{Opt}}$ when age is present versus age is discarded from the input data. Removing the age attribute slightly degrades the trade-off due to information discarding.

# 7 Conclusion

Invariant representation learning (IRepL) often involves a trade-off between utility and invariance. While the existence of such trade-off and its bounds have been studied, its *exact* characterization has not been investigated. This paper took steps towards addressing this problem by i) establishing the *exact* kernelized trade-off (denoted by K$-\mathcal{T}_{\text{Opt}}$), ii) determining the optimal dimensionality of the data representation necessary to achieve a desired optimal trade-off point, and iii) developing a scalable learning algorithm for encoders in some RKHSs to achieve K$-\mathcal{T}_{\text{Opt}}$. Numerical results on an illustrative example and two real-world datasets show that commonly used adversarial representation learning-based techniques cannot attain the optimal trade-off estimated by our solution.

Our theoretical results and empirical solutions shed light on the utility-invariance trade-off for various settings, such as algorithmic fairness and privacy-preserving learning under the scalarization of the bi-objective trade-off formulation. Furthermore, the trade-off in IRepL is also a function of the involved dependence measure that quantifies the dependence of learned representations on the semantic attribute. As such, the trade-off obtained in this paper is optimal for HSIC-like dependence measures. Studying the bi-objective trade-off (rather than the scalarization) and employing other universal measures are possible directions for future work.

## Broader Impact

IRepL can enable many machine learning systems to generalize to the domains that have not been trained on or prevent the leakage of private (sensitive) information while being effective for the desired prediction task(s). In particular, IRepL has a direct application in fairness which is a significant societal problem. Even though this paper aims to characterize the utility-invariance trade-off as a byproduct, our paper proposes an algorithm that learns fair representations of data. More generally, these approaches can enable machine learning systems to discard specific data before making predictions. We point out that demographic parity, the fairness criterion considered in this paper, can be unsuitable as a fairness criterion in some practical scenarios (Hardt et al., 2016; Chouldechova, 2017) and other fairness criteria like equalized odds (EO) or equality of opportunity (EOO) (Hardt et al., 2016) should be considered. The method proposed in this paper can be extended to other notions of fairness, such as EO and EOO, by modifying $\text{Dep}(Z, S)$ to capture the dependency between $Z$ and $S$, given $Y$. We leave this extension to future work.

**Acknowledgements:** This work was supported in part by financial assistance from the U.S. Department of Commerce, National Institute of Standards and Technology (award #60NANB18D210) and the National Science Foundation (award #2147116).

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

## A  A Population Expression for Definition in (7)

A population expression for $\mathrm{Dep}(Z, S)$ in (7) is given in the following.

$$\mathrm{Dep}(Z, S) = \sum_{j=1}^{r} \Big\{ \mathbb{E}_{X,S,X',S'} \left[ f_j(X) \, f_j(X') \, k_S(X, X') \right] + \mathbb{E}_X \left[ f_j(X) \right] \mathbb{E}_{X'} \left[ f_j(X') \right] \mathbb{E}_{S,S'} \left[ k_S(X, S') \right]$$

$$-2 \, \mathbb{E}_{X,S} \left[ f_j(X) \, \mathbb{E}_{X'}[f_j(X')] \, \mathbb{E}_{S'}[k_S(S, X')] \right] \Big\}$$

where $(X', S')$ is independent of $(X, S)$ with the same distribution as $p_{XS}$.

*Proof.* We first note that this population expression is inspired by that of HSIC (Gretton et al., 2005a).

Consider the operator $\Sigma_{SX}$ induced by the linear functional $\mathbb{Cov}(\alpha(X), \beta_S(S)) = \langle \beta_S, \Sigma_{SX} \alpha \rangle_{\mathcal{H}_S}$. Then, it follows that

$$\mathrm{Dep}(Z, S) = \sum_{j=1}^{r} \sum_{\beta_S \in \mathcal{U}_S} \mathbb{Cov}^2 \left( f_j(X), \beta_S(S) \right)$$

$$= \sum_{j=1}^{r} \sum_{\beta_S \in \mathcal{U}_S} \langle \beta_S, \Sigma_{SX} f_j \rangle_{\mathcal{H}_S}^2$$

$$= \sum_{j=1}^{r} \sum_{\beta_S \in \mathcal{U}_S} \langle \beta_S, \Sigma_{SX} f_j \rangle_{\mathcal{H}_S}^2$$

$$\overset{(a)}{=} \sum_{j=1}^{r} \| \Sigma_{SX} f_j \|_{\mathcal{H}_S}^2$$

$$= \sum_{j=1}^{r} \langle \Sigma_{SX} f_j, \Sigma_{SX} f_j \rangle_{\mathcal{H}_S}$$

$$\overset{(b)}{=} \sum_{j=1}^{r} \mathbb{Cov} \left( f_j(X), \, \left( \Sigma_{SX} f_j \right)(S) \right)$$

$$= \sum_{j=1}^{r} \mathbb{Cov} \left( f_j(X), \, \langle k_S(\cdot, S), \Sigma_{SX} f_j \rangle_{\mathcal{H}_S} \right)$$

$$= \sum_{j=1}^{r} \mathbb{Cov} \left( f_j(X), \mathbb{Cov}(f_j(X'), \, k_S(S', S)) \right)$$

$$= \sum_{j=1}^{r} \mathbb{Cov} \left( f_j(X), \, \mathbb{E}_{X',S'}[f_j(X') \, k_S(S, S')] - \mathbb{E}_{X'}[f_j(X')] \, \mathbb{E}_{S'}[k_S(S, S')] \right)$$

$$= \sum_{j=1}^{r} \Big\{ \mathbb{E}_{X,S,X',S'} \left[ f_j(X) \, f_j(X') \, k_S(S, S') \right] + \mathbb{E}_X \left[ f_j(X) \right] \mathbb{E}_{X'} \left[ f_j(X') \right] \mathbb{E}_{S,S'} \left[ k_S(S, S') \right]$$

$$-2 \, \mathbb{E}_{X,S} \left[ f_j(X) \, \mathbb{E}_{X'}[f_j(X')] \, \mathbb{E}_{S'}[k_S(S, S')] \right] \Big\}$$

where (a) is due to Parseval relation for orthonormal basis and (b) is from the definition of $\Sigma_{SX}$. $\qquad\square$

## B  Proof of Lemma 1

**Lemma 1.** Let $\boldsymbol{K}_X, \boldsymbol{K}_S \in \mathbb{R}^{n \times n}$ be Gram matrices corresponding to $\mathcal{H}_X$ and $\mathcal{H}_S$, respectively, i.e., $(\boldsymbol{K}_X)_{ij} = k_X(\boldsymbol{x}_i, \boldsymbol{x}_j)$ and $(\boldsymbol{K}_S)_{ij} = k_S(\boldsymbol{s}_i, \boldsymbol{s}_j)$, where covariance is empirically estimated as

$$\mathbb{Cov}(f_j(X), \beta_S(S)) \approx \frac{1}{n} \sum_{i=1}^{n} f_j(\boldsymbol{x}_i) \beta_S(\boldsymbol{s}_i) - \frac{1}{n^2} \sum_{i=1}^{n} \sum_{k=1}^{n} f_j(\boldsymbol{x}_i) \beta_S(\boldsymbol{s}_k).$$

It follows that, the corresponding empirical estimation for $\text{Dep}\,(Z, S)$ is

$$\text{Dep}^{\text{emp}}\,(Z, S) \quad := \quad \frac{1}{n^2}\,\|\boldsymbol{\Theta}\boldsymbol{K}_X\boldsymbol{H}\boldsymbol{L}_S\|_F^2\,, \tag{20}$$

where $\boldsymbol{H} = \boldsymbol{I}_n - \frac{1}{n}\boldsymbol{1}_n\boldsymbol{1}_n^T$ is the centering matrix, and $\boldsymbol{L}_S$ is a full column-rank matrix in which $\boldsymbol{L}_S\boldsymbol{L}_S^T = \boldsymbol{K}_S$ (Cholesky factorization), and $\boldsymbol{K}_S$ is the Gram matrix corresponding to $\mathcal{H}_S$. Furthermore, the empirical estimator in (9) has a bias of $\mathcal{O}(n^{-1})$ and a convergence rate of $\mathcal{O}(n^{-1/2})$.

*Proof.* Firstly, let us reconstruct the orthonormal set $\mathcal{U}_S$ when $n$ i.i.d. observations $\{\boldsymbol{s}_j\}_{j=1}^n$ are given. Invoking representer theorem, for two arbitrary elements $\beta_i$ and $\beta_m$ in $\mathcal{U}_S$, we have

$$
\begin{aligned}
\langle \beta_i,\, \beta_m \rangle_{\mathcal{H}_S} \quad &= \quad \left\langle \sum_{j=1}^n \alpha_j k_S(\boldsymbol{s}_j, \cdot),\, \sum_{l=1}^n \eta_l k_S(\boldsymbol{s}_l, \cdot) \right\rangle_{\mathcal{H}_S} \\
&= \quad \sum_{j=1}^n \sum_{l=1}^n \alpha_j \eta_l k_S(\boldsymbol{s}_j, \boldsymbol{s}_l) \\
&= \quad \boldsymbol{\alpha}^T \boldsymbol{K}_S \boldsymbol{\eta} \\
&= \quad \left\langle \boldsymbol{L}_S^T \boldsymbol{\alpha},\, \boldsymbol{L}_S^T \boldsymbol{\eta} \right\rangle_{\mathbb{R}^q}
\end{aligned}
$$

where $\boldsymbol{L}_S \in \mathbb{R}^{n \times q}$ is a full column-rank matrix and $\boldsymbol{K}_S = \boldsymbol{L}_S\boldsymbol{L}_S^T$ is the Cholesky factorization of $\boldsymbol{K}_S$. As a result, searching for $\beta_i \in \mathcal{U}_S$ is equivalent to searching for $\boldsymbol{L}_S^T\boldsymbol{\alpha} \in \mathcal{U}_q$ where $\mathcal{U}_q$ is any complete orthonormal set for $\mathbb{R}^q$. Using empirical expression for covariance, we get

$$
\begin{aligned}
\text{Dep}^{\text{emp}}(Z, S) \quad &:= \quad \sum_{\beta_S \in \mathcal{U}_S} \sum_{j=1}^r \left\{ \frac{1}{n}\sum_{i=1}^n f_j(\boldsymbol{x}_i)\beta_S(\boldsymbol{s}_i) - \frac{1}{n^2}\sum_{i=1}^n f_j(\boldsymbol{x}_i)\sum_{k=1}^n \beta_S(\boldsymbol{s}_k) \right\}^2 \\
&= \quad \sum_{\boldsymbol{L}_S^T\boldsymbol{\alpha} \in \mathcal{U}_q} \sum_{j=1}^r \left\{ \frac{1}{n}\boldsymbol{\theta}_j^T \boldsymbol{K}_X\boldsymbol{K}_S\boldsymbol{\alpha} - \frac{1}{n^2}\boldsymbol{\theta}_j^T\boldsymbol{K}_X\boldsymbol{1}_n\boldsymbol{1}_n^T\boldsymbol{K}_S\boldsymbol{\alpha} \right\}^2 \\
&= \quad \sum_{\boldsymbol{L}_S^T\boldsymbol{\alpha} \in \mathcal{U}_q} \sum_{j=1}^r \left\{ \frac{1}{n}\boldsymbol{\theta}_j^T \boldsymbol{K}_X\boldsymbol{H}\boldsymbol{K}_S\boldsymbol{\alpha} \right\}^2 \\
&= \quad \sum_{\boldsymbol{L}_S^T\boldsymbol{\alpha} \in \mathcal{U}_q} \sum_{j=1}^r \left\{ \frac{1}{n}\boldsymbol{\theta}_j^T \boldsymbol{K}_X\boldsymbol{H}\boldsymbol{L}_S\boldsymbol{L}_S^T\boldsymbol{\alpha} \right\}^2 \\
&= \quad \sum_{\boldsymbol{\zeta} \in \mathcal{U}_q} \sum_{j=1}^r \left\{ \frac{1}{n}\boldsymbol{\theta}_j^T \boldsymbol{K}_X\boldsymbol{H}\boldsymbol{L}_S\boldsymbol{\zeta} \right\}^2 \\
&= \quad \sum_{\boldsymbol{\zeta} \in \mathcal{U}_q} \frac{1}{n^2}\|\boldsymbol{\Theta}\boldsymbol{K}_X\boldsymbol{H}\boldsymbol{L}_S\boldsymbol{\zeta}\|_2^2 \\
&= \quad \frac{1}{n^2}\|\boldsymbol{\Theta}\boldsymbol{K}_X\boldsymbol{H}\boldsymbol{L}_S\|_F^2\,,
\end{aligned}
$$

where $\boldsymbol{f}(X) = \boldsymbol{\Theta}\big[k_X(\boldsymbol{x}_1, X), \cdots, k_X(\boldsymbol{x}_n, X)\big]^T$ and $\boldsymbol{\Theta} := [\boldsymbol{\theta}_1, \cdots, \boldsymbol{\theta}_r]^T$.

We now show that the bias of $\mathrm{Dep}^{\mathrm{epm}}(Z, S)$ for estimating $\mathrm{Dep}(Z, S)$ in (9) is $\mathcal{O}\left(\frac{1}{n}\right)$. To achieve this, we split $\mathrm{Dep}^{\mathrm{epm}}(Z, S)$ into three terms as,

$$
\begin{aligned}
\frac{1}{n^2}\left\|\boldsymbol{\Theta}\boldsymbol{K}_X\boldsymbol{H}\boldsymbol{L}_S\right\|_F^2 &= \frac{1}{n^2}\mathrm{Tr}\left\{\boldsymbol{\Theta}\boldsymbol{K}_X\boldsymbol{H}\boldsymbol{K}_S\boldsymbol{H}\boldsymbol{K}_X\boldsymbol{\Theta}^T\right\} \\
&= \frac{1}{n^2}\mathrm{Tr}\left\{\boldsymbol{\Theta}\boldsymbol{K}_X\left(\boldsymbol{I}-\frac{1}{n}\boldsymbol{1}\boldsymbol{1}^T\right)\boldsymbol{K}_S\left(\boldsymbol{I}-\frac{1}{n}\boldsymbol{1}\boldsymbol{1}^T\right)\boldsymbol{K}_X\boldsymbol{\Theta}^T\right\} \\
&= \frac{1}{n^2}\underbrace{\mathrm{Tr}\left\{\boldsymbol{K}_X\boldsymbol{\Theta}^T\boldsymbol{\Theta}\boldsymbol{K}_X\boldsymbol{K}_S\right\}}_{\mathrm{I}}-\frac{2}{n^3}\underbrace{\mathrm{Tr}\left\{\boldsymbol{1}^T\boldsymbol{K}_X\boldsymbol{\Theta}^T\boldsymbol{\Theta}\boldsymbol{K}_X\boldsymbol{K}_S\boldsymbol{1}\right\}}_{\mathrm{II}} \\
&\quad +\frac{1}{n^4}\underbrace{\mathrm{Tr}\left\{\boldsymbol{1}^T\boldsymbol{K}_X\boldsymbol{\Theta}^T\boldsymbol{\Theta}\boldsymbol{K}_X\boldsymbol{1}\boldsymbol{1}^T\boldsymbol{K}_S\boldsymbol{1}\right\}}_{\mathrm{III}}
\end{aligned}
\tag{21}
$$

Let $\boldsymbol{c}_p^n$ denote the set of all $p$-tuples drawn without replacement from $\{1,\cdots,n\}$. Moreover, let $\boldsymbol{\Theta}=[\boldsymbol{\theta}_1,\cdots,\boldsymbol{\theta}_r]^T \in \mathbb{R}^{r\times n}$ and $(\boldsymbol{A})_{ij}$ denote the element of an arbitrary matrix $\boldsymbol{A}$ at $i$-th row and $j$-th column. Then, it follows that

(I):

$$
\begin{aligned}
\mathbb{E}\left[\mathrm{Tr}\left\{\boldsymbol{K}_X\boldsymbol{\Theta}^T\boldsymbol{\Theta}\boldsymbol{K}_X\boldsymbol{K}_S\right\}\right] &= \sum_{k=1}^{r}\mathbb{E}\left[\mathrm{Tr}\left\{\underbrace{\boldsymbol{K}_X\boldsymbol{\theta}_k}_{:=\boldsymbol{\alpha}_k}\boldsymbol{\theta}_k^T\boldsymbol{K}_X\boldsymbol{K}_S\right\}\right] \\
&= \sum_{k=1}^{r}\mathbb{E}\left[\mathrm{Tr}\left\{\boldsymbol{\alpha}_k\boldsymbol{\alpha}_k^T\boldsymbol{K}_S\right\}\right] \\
&= \sum_{k=1}^{r}\mathbb{E}\left[\sum_i(\boldsymbol{\alpha}_k\boldsymbol{\alpha}_k^T)_{ii}(\boldsymbol{K}_S)_{ii}+\sum_{(i,j)\in\boldsymbol{c}_2^n}(\boldsymbol{\alpha}_k\boldsymbol{\alpha}_k^T)_{ij}(\boldsymbol{K}_S)_{ji}\right] \\
&= n\sum_{k=1}^{r}\mathbb{E}_{X,S}\left[f_k^2(X)k_S(S,S)\right]+\frac{n!}{(n-2)!}\sum_{k=1}^{r}\mathbb{E}_{X,S,X',S'}\left[f_k(X)f_k(X')k_S(S,S')\right] \\
&= \mathcal{O}(n)+\frac{n!}{(n-2)!}\sum_{k=1}^{r}\mathbb{E}_{X,S,X',S'}\left[f_k(X)f_k(X')k_S(S,S')\right]
\end{aligned}
\tag{22}
$$

where $(X, S)$ and $(X', S')$ are independently drawn from the joint distribution $\boldsymbol{p}_{XS}$.

(II):

$$
\begin{aligned}
\mathbb{E}\left[\boldsymbol{1}^T\boldsymbol{K}_X\boldsymbol{\Theta}^T\boldsymbol{\Theta}\boldsymbol{K}_X\boldsymbol{K}_S\boldsymbol{1}\right] &= \sum_{k=1}^{r}\mathbb{E}\left[\boldsymbol{1}^T\underbrace{\boldsymbol{K}_X\boldsymbol{\theta}_k}_{\boldsymbol{\alpha}_k}\boldsymbol{\theta}_k^T\boldsymbol{K}_X\boldsymbol{K}_S\boldsymbol{1}\right] \\
&= \sum_{k=1}^{r}\mathbb{E}\left[\boldsymbol{1}^T\boldsymbol{\alpha}_k\boldsymbol{\alpha}_k^T\boldsymbol{K}_S\boldsymbol{1}\right] \\
&= \sum_{k=1}^{r}\mathbb{E}\left[\sum_{m=1}^{n}\sum_{i=1}^{n}\sum_{j=1}^{n}(\boldsymbol{\alpha}_k\boldsymbol{\alpha}_k^T)_{mi}(\boldsymbol{K}_S)_{mj}\right] \\
&= \sum_{k=1}^{r}\mathbb{E}\left[\sum_i(\boldsymbol{\alpha}_k\boldsymbol{\alpha}_k^T)_{ii}(\boldsymbol{K}_S)_{ii}+\sum_{(m,j)\in\boldsymbol{c}_2^n}(\boldsymbol{\alpha}_k\boldsymbol{\alpha}_k^T)_{mm}(\boldsymbol{K}_S)_{mj}\right] \\
&\quad +\sum_{k=1}^{r}\mathbb{E}\left[\sum_{(m,i)\in\boldsymbol{c}_2^n}(\boldsymbol{\alpha}_k\boldsymbol{\alpha}_k^T)_{mi}(\boldsymbol{K}_S)_{mm}+\sum_{(m,j)\in\boldsymbol{c}_2^n}(\boldsymbol{\alpha}_k\boldsymbol{\alpha}_k^T)_{mj}(\boldsymbol{K}_S)_{mj}\right]
\end{aligned}
$$

$$+ \quad \sum_{k=1}^{r} \mathbb{E}\left[ \sum_{(m,i,j)\in \boldsymbol{c}_3^n} (\boldsymbol{\alpha}_k \boldsymbol{\alpha}_k^T)_{mi}(\boldsymbol{K}_S)_{mj} \right]$$

$$= \quad n\sum_{k=1}^{r} \mathbb{E}_{X,S}\left[ f_k^2(X)k_S(S,S) \right] + \frac{n!}{(n-2)!}\sum_{k=1}^{r}\mathbb{E}_{X,S,S'}\left[ f_k^2(X)k_S(S,S') \right]$$

$$+ \quad \frac{n!}{(n-2)!}\sum_{k=1}^{r}\mathbb{E}_{X,S,X'}\left[ f_k(X)f_k(X')k_S(S,S) \right]$$

$$+ \quad \frac{n!}{(n-2)!}\sum_{k=1}^{r}\mathbb{E}_{X,S,X',S'}\left[ f_k(X)f_k(X')k_S(S,S') \right]$$

$$+ \quad \frac{n!}{(n-3)!}\sum_{k=1}^{r}\mathbb{E}_{X,S}\left[ f_k(X)\mathbb{E}_{X'}[f_k(X')]\mathbb{E}_{S'}[k_S(S,S')] \right]$$

$$= \quad \mathcal{O}(n^2) + \frac{n!}{(n-3)!}\sum_{k=1}^{r}\mathbb{E}_{X,S}\left[ f_k(X)\mathbb{E}_{X'}[f_k(X')]\mathbb{E}_{S'}[k_S(S,S')] \right]. \tag{23}$$

(III):

$$\mathbb{E}\left[ \mathbf{1}^T \boldsymbol{K}_X \boldsymbol{\Theta}^T \boldsymbol{\Theta} \boldsymbol{K}_X \mathbf{1}\mathbf{1}^T \boldsymbol{K}_S \mathbf{1} \right] \quad = \quad \sum_{k=1}^{r}\mathbb{E}\left[ \mathbf{1}^T \underbrace{\boldsymbol{K}_X \boldsymbol{\theta}_k}_{\boldsymbol{\alpha}_k} \boldsymbol{\theta}_k^T \boldsymbol{K}_X \mathbf{1}\mathbf{1}^T \boldsymbol{K}_S \mathbf{1} \right]$$

$$= \quad \sum_{k=1}^{r}\mathbb{E}\left[ \mathbf{1}^T \boldsymbol{\alpha}_k \boldsymbol{\alpha}_k^T \mathbf{1}\mathbf{1}^T \boldsymbol{K}_S \mathbf{1} \right]$$

$$= \quad \sum_{k=1}^{r}\mathbb{E}\left[ \sum_{i,j,m,l} (\boldsymbol{\alpha}_k \boldsymbol{\alpha}_k^T)_{ij}(\boldsymbol{K}_S)_{ml} \right]$$

$$= \quad \mathcal{O}(n^3) + \sum_{k=1}^{r}\mathbb{E}\left[ \sum_{(i,j,m,l)\in \boldsymbol{c}_4^n} (\boldsymbol{\alpha}_k \boldsymbol{\alpha}_k^T)_{ij}(\boldsymbol{K}_S)_{ml} \right]$$

$$= \quad \mathcal{O}(n^3) + \frac{n!}{(n-4)!}\sum_{k=1}^{r}\mathbb{E}_X\left[ f_k(X) \right] E_{X'}\left[ f_k(X') \right] \mathbb{E}_{S,S'}\left[ k_S(S,S') \right]$$

$$\tag{24}$$

Using above calculations together with Lemma 2 lead to

$$\mathrm{Dep}(Z,S) = \mathbb{E}\left[ \mathrm{Dep}^{\mathrm{emp}}(Z,S) \right] + \mathcal{O}\left( \frac{1}{n} \right).$$

We now obtain the convergence of $\mathrm{dep}^{\mathrm{emp}}(Z,S)$. Consider the decomposition in (21) together with (22), (23), and (24). Let $\boldsymbol{\alpha}_k := \boldsymbol{K}_X \boldsymbol{\theta}_k$, then it follows that

$$\mathbb{P}\left\{ \mathrm{Dep}(Z,S) - \mathrm{Dep}^{\mathrm{emp}}(Z,S) \ge t \right\}$$

$$\le \quad \mathbb{P}\left\{ \sum_{k=1}^{r}\mathbb{E}_{X,S,X',S'}\left[ f_k(X)f_k(X')k_S(S,S') \right] - \frac{(n-2)!}{n!}\sum_{k=1}^{r}\sum_{(i,j)\in \boldsymbol{c}_2^n}(\boldsymbol{\alpha}_k \boldsymbol{\alpha}_k^T)_{ij}(\boldsymbol{K}_S)_{ji} + \mathcal{O}\left(\frac{1}{n}\right) \ge at \right\}$$

$$+ \quad \mathbb{P}\left\{ \sum_{k=1}^{r}\mathbb{E}_{X,S}\left[ f_k(X)\mathbb{E}_{X'}[f_k(X')]\mathbb{E}_{S'}[k_S(S,S')] \right] - \frac{(n-3)!}{n!}\sum_{k=1}^{r}\sum_{(i,j,m)\in \boldsymbol{c}_3^n}(\boldsymbol{\alpha}_k \boldsymbol{\alpha}_k^T)_{mi}(\boldsymbol{K}_S)_{mj} + \mathcal{O}\left(\frac{1}{n}\right) \ge bt \right\}$$

$$+ \quad \mathbb{P}\left\{ \sum_{k=1}^{r}E_X\left[ f_k(X) \right] E_{X'}\left[ f_k(X') \right] \mathbb{E}_{S,S'}\left[ k_S(S,S') \right] \right.$$

$$-\frac{(n-4)!}{n!}\sum_{k=1}^{r}\sum_{(i,j,m,l)\in \boldsymbol{c}_4^n}(\boldsymbol{\alpha}_k\boldsymbol{\alpha}_k^T)_{ij}(\boldsymbol{K}_S)_{ml}+\mathcal{O}\left(\frac{1}{n}\right)\geq (1-a-b)t\Bigg\},$$

where $a,b>0$ and $a+b<1$. For convenience, we omit the term $\mathcal{O}\left(\frac{1}{n}\right)$ and add it back in the last stage.

Define $\boldsymbol{\zeta}:=(X,S)$ and consider the following U-statistics (Hoeffding, 1994)

$$u_1(\boldsymbol{\zeta}_i,\boldsymbol{\zeta}_j)=\frac{(n-2)!}{n!}\sum_{(i,j)\in \boldsymbol{c}_2^n}\sum_{k=1}^{r}(\boldsymbol{\alpha}_k\boldsymbol{\alpha}_k^T)_{ij}(\boldsymbol{K}_S)_{ij}$$

$$u_2(\boldsymbol{\zeta}_i,\boldsymbol{\zeta}_j,\boldsymbol{\zeta}_m)=\frac{(n-3)!}{n!}\sum_{(i,j,m)\in \boldsymbol{c}_3^n}\sum_{k=1}^{r}(\boldsymbol{\alpha}_k\boldsymbol{\alpha}_k^T)_{mi}(\boldsymbol{K}_S)_{mj}$$

$$u_3(\boldsymbol{\zeta}_i,\boldsymbol{\zeta}_j,\boldsymbol{\zeta}_m,\boldsymbol{\zeta}_l)=\frac{(n-4)!}{n!}\sum_{(i,j,m,l)\in \boldsymbol{c}_4^n}\sum_{k=1}^{r}(\boldsymbol{\alpha}_k\boldsymbol{\alpha}_k^T)_{ij}(\boldsymbol{K}_S)_{ml}$$

Then, from Hoeffding's inequality (Hoeffding, 1994) it follows that

$$\mathbb{P}\left\{\text{Dep}(Z,S)-\text{Dep}^{\text{emp}}(Z,S)\geq t\right\}\leq e^{\frac{-2a^2t^2}{2r^2M^2}n}+e^{\frac{-2b^2t^2}{3r^2M^2}n}+e^{\frac{-2(1-a-b)^2t^2}{4r^2M^2}n},$$

where we assumed that $k_S(\cdot,\cdot)$ is bounded by one and $f_k^2(X_i)$ is bounded by $M$ for any $k=1,\cdots,r$ and $i=1,\cdots,n$. Further, if $0.22\leq a<1$, it holds that

$$e^{\frac{-2a^2t^2}{2r^2M^2}n}+e^{\frac{-2b^2t^2}{3r^2M^2}n}+e^{\frac{-2(1-a-b)^2t^2}{4r^2M^2}n}\leq 3e^{\frac{-a^2t^2}{r^2M^2}n}.$$

Consequently, we have

$$\mathbb{P}\left\{|\text{Dep}(Z,S)-\text{Dep}^{\text{emp}}(Z,S)|\geq t\right\}\leq 6e^{\frac{-a^2t^2}{r^2M^2}n}.$$

Therefore, with probability at least $1-\delta$, it holds

$$|\text{Dep}(Z,S)-\text{Dep}^{\text{emp}}(Z,S)|\leq \sqrt{\frac{r^2M^2\log(6/\sigma)}{\alpha^2 n}}+\mathcal{O}\left(\frac{1}{n}\right). \tag{25}$$

$\square$

## C  Proof of Theorem 2

**Theorem 2.** Let $Z=\boldsymbol{f}(X)$ be an arbitrary representation of the input data, where $\boldsymbol{f}\in\mathcal{H}_X$. Then, there exist an invertible Borel function $\boldsymbol{h}$, such that, $\boldsymbol{h}\circ\boldsymbol{f}$ belongs to $\mathcal{A}_r$.

*Proof.* Recall that the space of disentangled representation is

$$\mathcal{A}_r:=\left\{(f_1,\cdots,f_r)\;\middle|\;f_i,f_j\in\mathcal{H}_X,\;\mathbb{C}\text{ov}\left(f_i(X),f_j(X)\right)+\gamma\langle f_i,f_j\rangle_{\mathcal{H}_X}=\delta_{i,j}\right\},$$

where $\gamma>0$. Let $I_X$ denote the identity operator from $\mathcal{H}_X$ to $\mathcal{H}_X$. We claim that $\boldsymbol{h}:=[h_1,\cdots,h_r]$, where

$$\boldsymbol{G}_0=\begin{bmatrix}\langle f_1,f_1\rangle_{\mathcal{H}_X}&\cdots&\langle f_1,f_r\rangle_{\mathcal{H}_X}\\\vdots&\ddots&\vdots\\\langle f_r,f_1\rangle_{\mathcal{H}_X}&\cdots&\langle f_r,f_r\rangle_{\mathcal{H}_X}\end{bmatrix}$$

$$\boldsymbol{G}=\boldsymbol{G}_0^{-1/2}$$

$$h_j\circ\boldsymbol{f}=\sum_{m=1}^{r}g_{jm}\left(\Sigma_{XX}+\gamma I_X\right)^{-1/2}f_j,\quad\forall j=1,\cdots,r$$

is the desired invertible transformation. To see this, construct

$$
\begin{aligned}
& \mathbb{C}\mathrm{ov}\left(h_i(\boldsymbol{f}(X)), h_j(\boldsymbol{f}(X))\right) + \gamma \langle h_i \circ \boldsymbol{f}, h_j \circ \boldsymbol{f} \rangle_{\mathcal{H}_X} \\
= \quad & \langle h_i \circ \boldsymbol{f}, (\Sigma_{XX} + \gamma I_X) \, h_j \circ \boldsymbol{f} \rangle_{\mathcal{H}_X} \\
= \quad & \left\langle \sum_{m=1}^r g_{im} (\Sigma_{XX} + \gamma I_X)^{-1/2} f_i, \sum_{k=1}^r g_{jk} (\Sigma_{XX} + \gamma I_X)(\Sigma_{XX} + \gamma I_X)^{-1/2} f_j \right\rangle_{\mathcal{H}_X} \\
= \quad & \sum_{m=1}^r \sum_{k=1}^r g_{im} \, g_{jk} \, \langle f_i, f_j \rangle_{\mathcal{H}_X} = (\boldsymbol{G} \, \boldsymbol{G}_0 \, \boldsymbol{G})_{ij} = \delta_{i,j}
\end{aligned}
$$

The inverse of $\boldsymbol{h}$ is $\boldsymbol{h}' := [h'_1, \cdots, h'_r]$ where

$$
\boldsymbol{H} = \boldsymbol{G}_0^{1/2}
$$

$$
h'_j \circ \boldsymbol{h} = \sum_{m=1}^r h_{jm} (\Sigma_{XX} + \gamma I_X)^{1/2} h_j, \quad \forall j = 1, \cdots, r.
$$

$\square$

# D   Proof of Theorem 3

**Theorem 3.** Consider the operator $\Sigma_{SX}$ to be induced by the bi-linear functional $\mathbb{C}\mathrm{ov}(\alpha(X), \beta_S(S)) = \langle \Sigma_{SX} \alpha, \beta_S \rangle_{\mathcal{H}_S}$ and define $\Sigma_{YX}$ and $\Sigma_{XX}$, similarly. Then, a global optimizer for the optimization problem in (11) is the eigenfunctions corresponding to the $r$ largest eigenvalues of the following generalized eigenvalue problem

$$
\left((1-\lambda)\Sigma_{YX}^* \Sigma_{YX} - \lambda \Sigma_{SX}^* \Sigma_{SX}\right) \boldsymbol{f} = \tau \left(\Sigma_{XX} + \gamma I_X\right) \boldsymbol{f}, \tag{26}
$$

where $\gamma$ is the disentanglement regularization parameter defined in (10), and $\Sigma^*$ is the adjoint of $\Sigma$.

*Proof.* Consider $\mathrm{Dep}(Z, S)$ in (7):

$$
\begin{aligned}
\mathrm{Dep}(Z, S) \quad &= \quad \sum_{\beta_S \in \mathcal{U}_S} \sum_{j=1}^r \mathbb{C}\mathrm{ov}^2\left(f_j(X), \beta_S(S)\right) \\
&= \quad \sum_{j=1}^r \sum_{\beta_S \in \mathcal{U}_S} \langle \beta_S, \Sigma_{SX} f_j \rangle_{\mathcal{H}_S}^2 \\
&= \quad \sum_{j=1}^r \|\Sigma_{SX} f_j\|_{\mathcal{H}_S}^2,
\end{aligned}
$$

where the last step is due to Parseval's identity for orthonormal basis set. Similarly, we have $\mathrm{dep}(Z, Y) = \sum_{j=1}^r \|\Sigma_{YX} f_j\|_{\mathcal{H}_Y}^2$. Recall that $Z = \boldsymbol{f}(X) = [(f_1(X), \cdots, f_r(X)]$, then it follows that

$$
\begin{aligned}
J(\boldsymbol{f}(X)) \quad &= \quad (1-\lambda) \sum_{j=1}^r \|\Sigma_{YX} f_j\|_{\mathcal{H}_Y}^2 - \lambda \sum_{j=1}^r \|\Sigma_{SX} f_j\|_{\mathcal{H}_S}^2 \\
&= \quad (1-\lambda) \sum_{j=1}^r \langle \Sigma_{YX} f_j, \Sigma_{YX} f_j \rangle_{\mathcal{H}_Y} - \lambda \sum_{j=1}^r \langle \Sigma_{SX} f_j, \Sigma_{SX} f_j \rangle_{\mathcal{H}_S} \\
&= \quad \sum_{j=1}^r \langle f_j, \left((1-\lambda)\Sigma_{YX}^* \Sigma_{YX} - \lambda \Sigma_{SX}^* \Sigma_{SX}\right) f_j \rangle_{\mathcal{H}_X},
\end{aligned}
$$

where $\Sigma^*$ is the adjoint operator of $\Sigma$. Further, note that $\mathbb{C}\mathrm{ov}\left(f_i(X), f_j(X)\right)$ is equal to $\langle f_i, \Sigma_{XX} f_j \rangle_{\mathcal{H}_X}$. As a result, the optimization problem in (26) can be restated as

$$
\sup_{\langle f_i, (\Sigma_{XX} + \gamma I_X) f_k \rangle_{\mathcal{H}_X} = \delta_{i,k}} \sum_{j=1}^r \langle f_j, \left((1-\lambda)\Sigma_{YX}^* \Sigma_{YX} - \lambda \Sigma_{SX}^* \Sigma_{SX}\right) f_j \rangle_{\mathcal{H}_X}, \quad 1 \le i, k \le r
$$

where $I_X$ denotes identity operator from $\mathcal{H}_X$ to $\mathcal{H}_X$. This optimization problem is known as generalized Rayleigh quotient (Strawderman, 1999) and a possible solution to it is given by the eigenfunctions corresponding to the $r$ largest eigenvalues of the following generalized problem

$$((1 - \lambda)\Sigma_{XY}\Sigma_{YX} - \lambda\Sigma_{XS}\Sigma_{SX})\, f = \lambda\,(\Sigma_{XX} + \gamma I_X)\, f.$$

$\square$

## E   Proofs of Theorem 4 and Corollary 4.1

**Theorem 4.** Let the Cholesky factorization of $\boldsymbol{K}_X$ be $\boldsymbol{K}_X = \boldsymbol{L}_X\boldsymbol{L}_X^T$, where $\boldsymbol{L}_X \in \mathbb{R}^{n \times d}$ $(d \leq n)$ is a full column-rank matrix. Let $r \leq d$, then a solution to (13) is

$$\boldsymbol{f}^{\text{opt}}(X) = \boldsymbol{\Theta}^{\text{opt}}\,[k_X(\boldsymbol{x}_1, X), \cdots, k_X(\boldsymbol{x}_n, X)]^T$$

where $\boldsymbol{\Theta}^{\text{opt}} = \boldsymbol{U}^T\boldsymbol{L}_X^\dagger$ and the columns of $\boldsymbol{U}$ are eigenvectors corresponding to the $r$ largest eigenvalues of the following generalized eigenvalue problem.

$$\boldsymbol{L}_X^T\left((1 - \lambda)\boldsymbol{H}\boldsymbol{K}_Y\boldsymbol{H} - \lambda\boldsymbol{H}\boldsymbol{K}_S\boldsymbol{H}\right)\boldsymbol{L}_X\boldsymbol{u} = \tau\left(\frac{1}{n}\,\boldsymbol{L}_X^T\boldsymbol{H}\boldsymbol{L}_X + \gamma\boldsymbol{I}\right)\boldsymbol{u}. \tag{27}$$

Further, the supremum value of (13) is equal to $\sum_{j=1}^r \tau_j$, where $\{\tau_1, \cdots, \tau_r\}$ are $r$ largest eigenvalues of (14).

*Proof.* Consider the Cholesky factorization, $\boldsymbol{K}_X = \boldsymbol{L}_X\boldsymbol{L}_X^T$ where $\boldsymbol{L}_X$ is a full column-rank matrix. Using the representer theorem, the disentanglement property in (10) can be expressed as

$$
\begin{aligned}
&\mathbb{C}\text{ov}\left(f_i(X),\, f_j(X)\right) + \gamma\,\langle f_i, f_j\rangle_{\mathcal{H}_X} \\
=\ & \frac{1}{n}\sum_{k=1}^n f_i(\boldsymbol{x}_k)f_j(\boldsymbol{x}_k) - \frac{1}{n^2}\sum_{k=1}^n f_i(\boldsymbol{x}_k)\sum_{m=1}^n f_j(\boldsymbol{x}_m) + \gamma\,\langle f_i, f_j\rangle_{\mathcal{H}_X} \\
=\ & \frac{1}{n}\sum_{k=1}^n\sum_{t=1}^n \boldsymbol{K}_X(\boldsymbol{x}_k, \boldsymbol{x}_t)\theta_{it}\sum_{m=1}^n \boldsymbol{K}_X(\boldsymbol{x}_k, \boldsymbol{x}_m)\theta_{jm} - \frac{1}{n^2}\boldsymbol{\theta}_i^T\boldsymbol{K}_X\boldsymbol{1}_n\boldsymbol{1}_n^T\boldsymbol{K}_X\boldsymbol{\theta}_j + \gamma\,\langle f_i, f_j\rangle_{\mathcal{H}_X} \\
=\ & \frac{1}{n}\left(\boldsymbol{K}_X\boldsymbol{\theta}_i\right)^T\left(\boldsymbol{K}_X\boldsymbol{\theta}_j\right) - \frac{1}{n^2}\boldsymbol{\theta}_i^T\boldsymbol{K}_X\boldsymbol{1}_n\boldsymbol{1}_n^T\boldsymbol{K}_X\boldsymbol{\theta}_j + \gamma\left\langle\sum_{k=1}^n \theta_{ik}k_X(\cdot, \boldsymbol{x}_k),\, \sum_{t=1}^n \theta_{it}k_X(\cdot, \boldsymbol{x}_t)\right\rangle_{\mathcal{H}_X} \\
=\ & \frac{1}{n}\boldsymbol{\theta}_i^T\boldsymbol{K}_X\boldsymbol{H}\boldsymbol{K}_X\boldsymbol{\theta}_j + \gamma\,\boldsymbol{\theta}_i^T\boldsymbol{K}_X\boldsymbol{\theta}_j \\
=\ & \frac{1}{n}\boldsymbol{\theta}_i^T\boldsymbol{L}_X\left(\boldsymbol{L}_X^T\boldsymbol{H}\boldsymbol{L}_X + n\gamma\,\boldsymbol{I}\right)\boldsymbol{L}_X^T\boldsymbol{\theta}_j \\
=\ & \delta_{i,j}.
\end{aligned}
$$

As a result, $\boldsymbol{f} \in \mathcal{A}_r$ is equivalent to

$$\boldsymbol{\Theta}\boldsymbol{L}_X\underbrace{\left(\frac{1}{n}\boldsymbol{L}_X^T\boldsymbol{H}\boldsymbol{L}_X + \gamma\boldsymbol{I}\right)}_{:=\boldsymbol{C}}\boldsymbol{L}_X^T\boldsymbol{\Theta}^T = \boldsymbol{I}_r,$$

where $\boldsymbol{\Theta} := \left[\boldsymbol{\theta}_1, \cdots, \boldsymbol{\theta}_r\right]^T \in \mathbb{R}^{r \times n}$.

Let $\boldsymbol{V} = \boldsymbol{L}_X^T \boldsymbol{\Theta}^T$ and consider the optimization problem in (13):

$$
\sup_{\boldsymbol{f} \in \mathcal{A}_r} \left\{ (1 - \lambda) \operatorname{Dep}^{\mathrm{emp}}(\boldsymbol{f}(X), Y) - \lambda \operatorname{Dep}^{\mathrm{emp}}(\boldsymbol{f}(X), S) \right\}
$$

$$
= \sup_{\boldsymbol{f} \in \mathcal{A}_r} \frac{1}{n^2} \left\{ (1 - \lambda) \|\boldsymbol{\Theta} \boldsymbol{K}_X \boldsymbol{H} \boldsymbol{L}_Y\|_F^2 - \lambda \|\boldsymbol{\Theta} \boldsymbol{K}_X \boldsymbol{H} \boldsymbol{L}_S\|_F^2 \right\}
$$

$$
= \sup_{\boldsymbol{f} \in \mathcal{A}_r} \frac{1}{n^2} \left\{ (1 - \lambda) \operatorname{Tr} \left\{ \boldsymbol{\Theta} \boldsymbol{K}_X \boldsymbol{H} \boldsymbol{K}_Y \boldsymbol{H} \boldsymbol{K}_X \boldsymbol{\Theta}^T \right\} - \lambda \operatorname{Tr} \left\{ \boldsymbol{\Theta} \boldsymbol{K}_X \boldsymbol{H} \boldsymbol{K}_S \boldsymbol{H} \boldsymbol{K}_X \boldsymbol{\Theta}^T \right\} \right\}
$$

$$
= \max_{\boldsymbol{V}^T \boldsymbol{C} \boldsymbol{V} = \boldsymbol{I}_r} \frac{1}{n^2} \operatorname{Tr} \left\{ \boldsymbol{\Theta} \boldsymbol{L}_X \boldsymbol{B} \boldsymbol{L}_X^T \boldsymbol{\Theta}^T \right\}
$$

$$
= \max_{\boldsymbol{V}^T \boldsymbol{C} \boldsymbol{V} = \boldsymbol{I}_r} \frac{1}{n^2} \operatorname{Tr} \left\{ \boldsymbol{V}^T \boldsymbol{B} \boldsymbol{V} \right\} \tag{28}
$$

where the second step is due to (9) and

$$
\boldsymbol{B} \quad := \quad \boldsymbol{L}_X^T \left( (1 - \lambda) \boldsymbol{H} \boldsymbol{K}_Y \boldsymbol{H} - \lambda \boldsymbol{H} \boldsymbol{K}_S \boldsymbol{H} \right) \boldsymbol{L}_X
$$

It is shown in Kokiopoulou et al. (2011) that an[3] optimizer of (28) is any matrix $\boldsymbol{U}$ whose columns are eigenvectors corresponding to $r$ largest eigenvalues of generalized problem

$$
\boldsymbol{B} \boldsymbol{u} = \tau \boldsymbol{C} \boldsymbol{u} \tag{29}
$$

and the maximum value is the summation of $r$ largest eigenvalues. Once $\boldsymbol{U}$ is determined, then, any $\boldsymbol{\Theta}$ in which $\boldsymbol{L}_X^T \boldsymbol{\Theta}^T = \boldsymbol{U}$ is optimal $\boldsymbol{\Theta}$ (denoted by $\boldsymbol{\Theta}^{\mathrm{opt}}$). Note that $\boldsymbol{\Theta}^{\mathrm{opt}}$ is not unique and has a general form of

$$
\boldsymbol{\Theta}^T = \left( \boldsymbol{L}_X^T \right)^\dagger \boldsymbol{U} + \boldsymbol{\Lambda}_0, \quad \mathcal{R}(\boldsymbol{\Lambda}_0) \subseteq \mathcal{N} \left( \boldsymbol{L}_X^T \right).
$$

However, setting $\boldsymbol{\Lambda}_0$ to zero would lead to minimum norm for $\boldsymbol{\Theta}$. Therefore, we opt $\boldsymbol{\Theta}^{\mathrm{opt}} = \boldsymbol{U}^T \boldsymbol{L}_X^\dagger$. $\qquad \square$

**Corollary 4.1.** *Embedding Dimensionality*: A useful corollary of Theorem 4 is characterizing optimal embedding dimensionality as a function of trade-off parameter, $\lambda$:

$$
r^{\mathrm{Opt}}(\lambda) := \arg \sup_{0 \leq r \leq l} \left\{ \sup_{\boldsymbol{f} \in \mathcal{A}_r} \left\{ J^{\mathrm{emp}}(\boldsymbol{f}, \lambda) \right\} \right\} = \text{ number of non-negative eigenvalues of (14)}
$$

*Proof.* From proof of Theorem 4, we know that

$$
\sup_{\boldsymbol{f} \in \mathcal{A}_r} \left\{ (1 - \lambda) \operatorname{Dep}^{\mathrm{emp}}(\boldsymbol{f}(X), Y) - \lambda \operatorname{Dep}^{\mathrm{emp}}(\boldsymbol{f}(X), S) \right\} = \sum_{j=1}^{r} \tau_j,
$$

where $\{\tau_1, \cdots, \tau_n\}$ are eigenvalues of the generalized problem in (14) in decreasing order. It follows immediately that

$$
\arg \sup_r \left\{ \sum_{j=1}^{r} \tau_j \right\} = \text{ number of non-negative elements of } \{\tau_1, \cdots, \tau_l\}.
$$

$\qquad \square$

# F  Proof of Theorem 5

**Theorem 5.** Assume that $k_S$ and $k_Y$ are bounded by one and $f_j^2(\boldsymbol{x}_i) \leq M$ for any $j = 1, \ldots, r$ and $i = 1, \ldots, n$ for which $\boldsymbol{f} = (f_1, \ldots, f_r) \in \mathcal{A}_r$. Then, for any $n > 1$ and $0 < \delta < 1$, with probability at least $1 - \delta$, we have

$$
\left| \sup_{\boldsymbol{f} \in \mathcal{A}_r} J(\boldsymbol{f}, \lambda) - \sup_{\boldsymbol{f} \in \mathcal{A}_r} J^{\mathrm{emp}}(\boldsymbol{f}, \lambda) \right| \leq r M \sqrt{\frac{\log(6/\delta)}{0.22^2 \, n}} + \mathcal{O}\left( \frac{1}{n} \right).
$$

---

[3]Optimal $\boldsymbol{V}$ is not unique.

*Proof.* Recall that in the proof of Lemma 1 we have shown that with probability at least $1 - \delta$, the following inequality holds

$$|\text{Dep}(Z, S) - \text{Dep}^{\text{emp}}(Z, S)| \leq \sqrt{\frac{r^2 M^2 \log(6/\sigma)}{0.22^2 \, n}} + \mathcal{O}\left(\frac{1}{n}\right).$$

Using the same reasoning for $\text{dep}(Z, Y)$, with probability at least $1 - \delta$, we have

$$|\text{Dep}(Z, Y) - \text{Dep}^{\text{emp}}(Z, Y)| \leq \sqrt{\frac{r^2 M^2 \log(6/\sigma)}{0.22^2 \, n}} + \mathcal{O}\left(\frac{1}{n}\right).$$

Since $J(\boldsymbol{f}(X)) = (1 - \lambda) \text{dep}(Z, Y) - \lambda \text{dep}(Z, S)$ and $J^{\text{emp}}(\boldsymbol{f}(X)) := (1 - \lambda) \text{dep}^{\text{emp}}(Z, Y) - \lambda \text{dep}^{\text{emp}}(Z, S)$, it follows that with probability at least $1 - \delta$,

$$|J(\boldsymbol{f}, \lambda) - J^{\text{emp}}(\boldsymbol{f}, \lambda)| \leq rM\sqrt{\frac{\log(6/\sigma)}{0.22^2 \, n}} + \mathcal{O}\left(\frac{1}{n}\right).$$

We complete the proof by noting that, the following inequality holds for any bounded $J$ and $J^{\text{emp}}$:

$$\left| \sup_{\boldsymbol{f} \in \mathcal{A}_r} J(\boldsymbol{f}, \lambda) - \sup_{\boldsymbol{f} \in \mathcal{A}_r} J^{\text{emp}}(\boldsymbol{f}, \lambda) \right| \leq \sup_{\boldsymbol{f} \in \mathcal{A}_r} |J(\boldsymbol{f}, \lambda) - J^{\text{emp}}(\boldsymbol{f}, \lambda)|.$$

$\square$

# G   Optimality of Target Task Performance in $\mathbf{K} - \mathcal{T}_{\mathbf{Opt}}$

We show that maximizing $\text{dep}(\boldsymbol{f}(X), Y)$ can lead to a representation $Z$ that is sufficient to result in the optimal Bayes prediction of $Y$.

**Theorem 6.** Let $\boldsymbol{f}^*$ be the optimal encoder by maximizing $\text{Dep}(\boldsymbol{f}(X), Y)$, where $\gamma \to 0$ and $\mathcal{H}_Y$ is a linear RKHS. Then, there exist $\boldsymbol{W} \in \mathbb{R}^{d_Y \times r}$ and $\boldsymbol{b} \in \mathbb{R}^{d_Y}$ such that $\boldsymbol{W}\boldsymbol{f}^*(X) + \boldsymbol{b}$ is the Bayes estimator, i.e.,

$$\mathbb{E}_{X,Y}\left[\|\boldsymbol{W}\boldsymbol{f}(X)^* + \boldsymbol{b} - Y\|^2\right] = \inf_{h \text{ is Borel}} \mathbb{E}_{X,Y}\left[\|h(X) - Y\|^2\right]$$
$$= \mathbb{E}_{X,Y}\left[\|\mathbb{E}[Y \,|\, X] - Y\|^2\right].$$

*Proof.* We only prove this theorem for the empirical version due to its convergence to the population counterpart. The optimal Bayes estimator can be the composition of the kernelized encoder $Z = \boldsymbol{f}(X)$ and a linear regressor on top of it. More specifically, $\widehat{Y} = \boldsymbol{W}\boldsymbol{f}(X) + \boldsymbol{b}$ can approach to $\mathbb{E}[Y \,|\, X]$ if we optimize $\boldsymbol{f}$, $\boldsymbol{W}$, and $\boldsymbol{b}$ all together. This is because $\boldsymbol{f} \in \mathcal{H}_X$ can approximate any Borel function (due to the universality of $\mathcal{H}_X$) and, since $r \geq d_y$, $\boldsymbol{W}$ can be surjective. Let $\boldsymbol{Z} := [\boldsymbol{z}_1, \cdots, \boldsymbol{z}_n] \in \mathbb{R}^{r \times n}$ and $\boldsymbol{Y} := [\boldsymbol{y}_1, \cdots, \boldsymbol{y}_n] \in \mathbb{R}^{d_y \times n}$. Further, let $\tilde{\boldsymbol{Z}}$ and $\tilde{\boldsymbol{y}}$ be the centered (i.e., mean subtracted) version of $\boldsymbol{Z}$ and $\boldsymbol{Y}$, respectively. We firstly optimize $\boldsymbol{b}$ for any given $\boldsymbol{f}$, $r$, and $\boldsymbol{W}$:

$$\boldsymbol{b}_{\text{opt}} := \arg\min_{\boldsymbol{b}} \frac{1}{n} \sum_{i=1}^{n} \|\boldsymbol{W}\boldsymbol{z}_i + \boldsymbol{b} - \boldsymbol{y}_i\|^2$$
$$= \frac{1}{n} \sum_{i=1}^{n} \boldsymbol{y}_i - \boldsymbol{W}\frac{1}{n} \sum_{i=1}^{n} \boldsymbol{z}_i.$$

Then, optimizing over $\boldsymbol{W}$ would lead to

$$\min_{\boldsymbol{W}} \frac{1}{n} \left\|\boldsymbol{W}\tilde{\boldsymbol{Z}} - \tilde{\boldsymbol{Y}}\right\|_F^2 = \frac{1}{n} \min_{\boldsymbol{W}} \left\|\tilde{\boldsymbol{Z}}^T \boldsymbol{W}^T - \tilde{\boldsymbol{Y}}^T\right\|_F^2$$
$$= \min_{\boldsymbol{W}} \frac{1}{n} \left\|\tilde{\boldsymbol{Z}}^T \boldsymbol{W}^T - P_{\tilde{\boldsymbol{Z}}}\tilde{\boldsymbol{Y}}^T\right\|_F^2 + \frac{1}{n} \left\|P_{\tilde{\boldsymbol{Z}}^\perp}\tilde{\boldsymbol{Y}}^T\right\|_F^2$$
$$= \frac{1}{n} \left\|P_{\tilde{\boldsymbol{Z}}^\perp}\tilde{\boldsymbol{Y}}^T\right\|_F^2 = \frac{1}{n} \left\|\tilde{\boldsymbol{Y}}\right\|_F^2 - \frac{1}{n} \left\|P_{\tilde{\boldsymbol{Z}}}\tilde{\boldsymbol{Y}}^T\right\|_F^2,$$

where $P_{\tilde{Z}}$ denotes the orthogonal projector onto the column space of $\tilde{Z}^T$ and a possible minimizer is $W_{\text{opt}}^T = (\tilde{Z}^T)^\dagger \tilde{Y}^T$ or equivalently $W_{\text{opt}} = \tilde{Y}(\tilde{Z})^\dagger$. Since the MSE loss is a function of the range (column space) of $\tilde{Z}^T$, we can consider only $\tilde{Z}^T$ with orthonormal columns or equivalently $\frac{1}{n}\tilde{Z}\tilde{Z}^T = I_r$. In this setting, it holds $P_{\tilde{Z}} = \frac{1}{n}\tilde{Z}^T\tilde{Z}$. Now, consider optimizing $f(X) = \Theta\left[k_X(x_1, X), \cdots, k_X(x_n, X)\right]^T$. We have, $\tilde{Z} = \Theta K_X H$ where $H$ is the centering matrix. Let $V = L_x^T\Theta^T$ and $C = \frac{1}{n}L_X^THL_X$, then it follows that

$$
\begin{aligned}
\min_{\Theta K_X H K_X \Theta^T = nI_r} \frac{1}{n}\left\{\|\tilde{Y}\|_F^2 - \|P_{\tilde{Z}}\tilde{Y}^T\|_F^2\right\} &= \frac{1}{n}\|\tilde{Y}\|_F^2 - \max_{\Theta K_X H K_X \Theta^T = nI_r}\frac{1}{n}\|P_{\tilde{Z}}\tilde{Y}^T\|_F^2 \\
&= \frac{1}{n}\|\tilde{Y}\|_F^2 - \max_{V^TCV=I_r}\frac{1}{n^2}\text{Tr}\left[\tilde{Y}HK_X\Theta^T\Theta K_X H\tilde{Y}^T\right] \\
&= \frac{1}{n^2}\|\tilde{Y}\|_F^2 - \max_{V^TCV=I_r}\frac{1}{n^2}\text{Tr}\left[\Theta K_X H\tilde{Y}^T\tilde{Y}HK_X\Theta^T\right] \\
&= \|\tilde{Y}\|_F^2 - \max_{V^TCV=I_r}\frac{1}{n^2}\text{Tr}\left[V^TL_X^T\tilde{Y}^T\tilde{Y}L_XV\right] \\
&= \frac{1}{n}\|\tilde{Y}\|_F^2 - \frac{1}{n^2}\sum_{j=1}^{r}\lambda_j,
\end{aligned}
$$

where $\lambda_1, \cdots, \lambda_r$ are $r$ largest eigenvalues of the following generalized problem

$$B_0 u = \lambda C u$$

and $B_0 := L_X^T\tilde{Y}^T\tilde{Y}L_X$. This resembles the eigenvalue problem in Section E, equation (29) where $\lambda = 0$, $\mathcal{H}_Y$ is a linear RKHS and $\gamma \to 0$. $\qquad\square$

## H Deficiency of Mean-Squared Error as A Measure of Dependence

**Theorem.** Let $\mathcal{H}_S$ contain all Borel functions, $S$ be a $d_S$-dimensional RV, and $L_S(\cdot, \cdot)$ be MSE loss. Then,

$$Z \in \arg\sup\left\{\inf_{g_S \in \mathcal{H}_S}\mathbb{E}_{X,S}\left[L_S\left(g_S\left(Z\right), S\right)\right]\right\} \Leftrightarrow \mathbb{E}[S\,|\,Z] = \mathbb{E}[S].$$

*Proof.* Let $S_i$, $(g_S(Z))_i$, and $(\mathbb{E}[S\,|\,Z])_i$ denote the $i$-th entries of $S$, $g_S(Z)$, and $\mathbb{E}[S\,|\,Z]$, respectively. Then, it follows that

$$
\begin{aligned}
\inf_{g_S \in \mathcal{H}_S}\mathbb{E}_{X,S}\left[L_S\left(g_S\left(Z\right), S\right)\right] &= \inf_{g_S \in \mathcal{H}_S}\sum_{i=1}^{d_S}\mathbb{E}_{X,S}\left[\left((g_S(Z))_i - S_i\right)^2\right] \\
&= \sum_{i=1}^{d_S}\mathbb{E}_{X,S}\left[\left((\mathbb{E}[S\,|\,Z])_i - S_i\right)^2\right] \\
&\leq \sum_{i=1}^{d_S}\mathbb{E}_S\left[\left((\mathbb{E}[S])_i - S_i\right)^2\right] = \sum_{i=1}^{d_S}\mathbb{V}\text{ar}[S_i],
\end{aligned}
$$

where the second step is due to the optimality of conditional mean (i.e., Bayes estimation) for MSE (Jacod & Protter, 2012) and the last step is because independence between $Z$ and $S$ leads to an upper bound on MSE. Therefore, if $Z \in \arg\sup\{\inf_{g_S \in \mathcal{H}_S}\mathbb{E}_{X,S}\left[L_S\left(g_S\left(Z\right), S\right)\right]\}$, then $\mathbb{E}[S\,|\,Z] = \mathbb{E}[S]$. On the other hand, if $\mathbb{E}[S\,|\,Z] = \mathbb{E}[S]$, then it follows immediately that $Z \in \arg\sup\{\inf_{g_S \in \mathcal{H}_S}\mathbb{E}_{X,S}\left[L_S\left(g_S\left(Z\right), S\right)\right]\}$. $\qquad\square$

This theorem implies that an optimal adversary does not necessarily lead to a representation $Z$ that is statistically independent of $S$, but rather leads to $S$ being mean independent of the representation $Z$.

