# OpenReview forum: "On Characterizing the Trade-off in Invariant Representation Learning"
_TMLR — Accepted by TMLR_

### Review · Reviewer_LPRL · 2022-10-18

**Summary Of Contributions:**

In its seminal form, the Invariance Representation Learning (IRL) consists of finding an encoder $f$ and a model $g$ that would minimize
$$ L(g(f(X)), Y) $$
under a constraint $D(f(X), S) < \epsilon$, where
- $(X,Y,S)$ is a joint distribution of observations, labels and semantic attributes (e.g. $X$ is some person, $S$ their gender, and $Y$ an math score for a job offer).
- Minimization is understood in expectation w.r.t. this joint distribution
- $D$ is a correlation-like loss measuring the dependency between the representation $Z = f(X)$ and the semantic attribute $S$, and $\epsilon$ a control parameter. Typically, setting $\epsilon = 0$ would enforce that the representation $f(X)$ (hence the model $g \circ f (X)$) is independent of the semantic attribute $S$.

While interesting in itself, this problem is hard to handle from both an analytic and computational perspective. In order to overcome this issue, the authors consider a different (though similar model), that eventually reads

$$\sup_{f_1,\dots,f_r \in A_r} (1 - \lambda) D(f(X), Y) - \lambda D(f(X), S), $$

where
- $\lambda \in (0,1)$ is a tradeoff parameter
- $D(Z, S) = \sum_{j=1}^r \sum_\beta \mathrm{Cov}(f_j(Z), \beta(S))$, where $r$ is (roughly) an embedding dimension (that can be chosen optimally), the $\beta$s wander an orthonormal basis of a universal RKHS,
- $A_r$ denotes a space of functions $(f_1,\dots, f_r)$ that encourage the disantanglement of the $(Z_j)_{j=1}^r$ (through an additional parameter $\gamma$). The $f_j$ belong to an RKHS.

Therefore, the proposer model tends to learn a representation $Z = f(X)$ that has roughly independent coordinates, that is correlated to $Y$ while being independent of $S$, as much as possible.

The authors demonstrate the practicability of their formalism, thanks to the representer theorem in RKHS, most expression boils down to matrix manipulation and a solution of their problem can be estimated by solving an eigenvalue problem (Thm 4).
The authors showcase their approach through different numerical experiments on synthetic and real-life datasets.


**Audience:**

Yes

**Broader Impact Concerns:**

From my understanding, the Invariant Representation Learning field is intrinsically related to ethical question, as (one of) its goal is to design models that can be guaranteed to be insensitive (to some extend) to attribute such as gender, age, etc.
I think this work proposes an improvement in this direction and may have some impact in fairness/ethic-ML etc. (though I'm not an expert on those topics either). Perhaps a specific paragraph on this kind of question may be worth it (though the related work section is already fairly instructive on that matter).

**Claims And Evidence:**

Yes

**Requested Changes:**

I do not feel qualified to request major changes.

On the minor remark / clarification side :

- In Eq. (1), in which sense the inequality $D(f(X), S) \leq \epsilon$ should hold? Is this a.e., or in expectation ?
- In Eq. (2), shouln't the right braket $]$ in $\mathbb{E}_{XY}$ end after the term $D(f(X), S)$? And also, shouldn't the expectation be taken with respect to the join law $(X, Y, S)$ and not only $(X,Y)$ ?

**Strengths And Weaknesses:**

First, I stress that I am completely new to the IRL problem and this review is an educated guess.

# Strengths :

- The paper is well-written from my perspective, in sense that it introduces the problem in an understandable way for the non-expert, and the different steps are motivated.
- From what I can tell, the experimental section is fairly expanded and validate the proposed approach in various ways.
- While I did not proofread the appendices in details, proofs seem well-written and I did not identify a major flaw at first glance.

# Weaknesses :
- As far as I understand, the proposed model deviates quite significantly from the initial IRL problem. While each step is motivated, and the result satisfactory, I cannot assess if the resulting problem is, or is not, "too simplified" or "too different". In particular, I guess that using the same type of loss for the terms $(Z,Y)$ and $(Z,S)$ is important for the tractability of the solution and I cannot assess whether this is a reasonable setting or not.

---

> ### Author Response · Authors · 2022-10-29
> **Reply to Reviewer LPRL**
>
> Please see our changes in the revised manuscript in blue.
>
> > As far as I understand, the proposed model deviates quite significantly from the initial IRL problem...
>
> To the best of our knowledge, the original problem of the optimal trade-off ($\mathcal T_{\text{Opt}}$) in equation (2) cannot be solved optimally in closed form or algorithmically. Therefore, no accurate/reliable utility-invariance trade-off can be obtained.  Even though we agree that the problem is simplified and altered compared to the original definition of $\mathcal T_{\text{Opt}}$, however, the altered problem remains reasonably related to the original IRepL problem as discussed and motivated in Sections 3.2 and 5.2. In Section 3.2, we cited (Barshan et al., 2011) for using Dep$(Z, Y)$-like loss function for classification, regression, and clustering tasks. In Section 5.2, we theoretically demonstrate that by appropriate choice of the involved RKHS in Dep$(Z, Y)$, Dep$(Z, Y)$ can perform as optimally as Bayes estimator $\mathbb E[Y|X]$ for the regression task (i.e., MSE target loss).
>
>
> > In Eq. (1), in which sense the inequality $D(f(X), S) \leq \epsilon$ should hold? Is this a.e., or in expectation?
>
> It is in expectation. For example, see our definition for Dep$(f(X), S)$ in equation (7) where covariance is employed (i.e., an implicit expectation $\mathbb E_{X, S}$) to obtain Dep$(f(X), S)$. Therefore, one can assume that Dep$(f(X), S)$ is a deterministic quantity.
>
>
>
> > In Eq. (2), shouldn't the right bracket $]$ in $\mathbb{E}_{XY}$ end after the term $D(f(X), S)$? And also, shouldn't the expectation be taken with respect to the joint law $(X, Y, S)$ and not only $(X, Y)$?
>
> Please note that as we mentioned earlier, Dep$(f(X), S)$ is a deterministic value due to the implicit employment of $\mathbb E_{X, S}$. Moreover, the term $L_Y(g_Y(f(X)), Y)$ depends only on the distribution of $(X, Y)$.
>
>
>
> > Broader Impact Concerns: From my understanding, the Invariant Representation Learning field is ...
>
> We added the "Broader Impact" Section after the "Conclusion" Section where we discussed the societal impact of this work in fairness.

---

### Review · Reviewer_tLLR · 2022-10-18

**Summary Of Contributions:**

This paper studies the tradeoff between utility and independence when learning representations for classification/regression tasks. Given a dataset $X$, the goal is to learn a representation $f(X)$ which predicts a target $Y$ while remaining invariant w.r.t attributes $S$. By assuming the encoder belongs to a reproducing kernel Hilbert space (RKHS) and defining a modified dependence measure based on Hilbert-Schmidt Independence Criterion (HSIC), the authors characterize an exact closed-form tradeoff between these two competing desiderata. The paper then describes an efficient, approximate implementation and shows that the proposed method attains a better empirical tradeoff than previous approaches on synthetic and real-world datasets.

**Audience:**

Yes

**Broader Impact Concerns:**

No major concerns, but the authors could comment on the distinction between characterizing tradeoffs and designing models/datasets to obtain specific invariance/fairness properties

**Claims And Evidence:**

Yes

**Requested Changes:**

### Critical
- Include experiments/figures in Section 6.6 that vary $r$ rather than setting $r=1$ or $r=r^{Opt}(0)$
- Include proof sketches in the main text
- Implementation details: describe how neural architectures for ARL were selected. Are the number of network parameters comparable to the number of kernel parameters for $K\mbox{-}\mathcal{T}_{Opt}$?
- In addition to Figures 3-4, since Equation (11) characterizes the tradeoff between $Dep(Z, Y)$ and $Dep(Z, S)$ it would be good to include this tradeoff in Section 6.6

### Non-Critical
- Renaming the Invariant Representation Learning optimization problem may help avoid confusion with Inverse Reinforcement Learning, which uses the same acronym.
- If possible, it would be good to calculate theoretical upper/lower bounds for this experimental setup, and include them in Figures 3-4
- Include additional experiments on applications other than demographic parity (e.g. privacy, multi-domain generalization)

### Typos
- "reverse of a non-parametric measure" should be "negative of a non-parametric measure"
- "performs as optimum as" should be "performs as optimally as"


**Strengths And Weaknesses:**

### Strengths
- Good descriptions of/comparisons to related work, including how the proposed method differs from them
- Elegant, principled formulation
- Clearly written and generally well organized

### Weaknesses
- Some details in the experimental setup are missing and/or nonstandard. If $r^{Opt}(\lambda)$ provides a better tradeoff than $r^{Opt}(0)$ (shown in Section 6.7, Figure 5a), then the experiments/figures in Section 6.6 should also use the former. Additionally, setting $r=1$ does not capture the full complexity of $K\mbox{-}\mathcal{T}_{Opt}$ ; for example Equation (10) becomes trivial. The authors should confirm that empirical findings on real datasets still hold when $r>1$.
- Organization: The paper's presentation would be improved greatly if proof sketches/methods were included in the main text (e.g. Hoeffding's inequality, generalized Rayleigh quotient)
- The Introduction motivates invariant representation learning with multiple applications (privacy, multi-domain generalization). However, real-world experiments only focus on the use case of demographic parity.

---

> ### Author Response · Authors · 2022-10-29
> **Reply to Reviewer tLLR**
>
> > - Renaming the Invariant Representation Learning optimization problem may help avoid confusion with Inverse Reinforcement Learning, which uses the same acronym.
>
> We renamed IRL to IRepL to avoid confusion with Inverse Reinforcement Learning. Our changes are in blue in the revised manuscript.
>
>
> > - Include experiments/figures in Section 6.6 that vary $r$ rather than setting $r=1$ or $r=r^{\text{Opt}}(0)$
>
> We mentioned in Section 6.5 (under **Embedding Dimensionality** Subsection): "For K-$\mathcal T_{\text{Opt}}$, we use $r^{\text{Opt}}(\lambda)$". Please note that the blue curve in Figure 4 (a) is corresponding to $r^{\text{Opt}}(\lambda)$ not $r^{\text{Opt}}(0)$. We included a new experiment on New York (2018) census data where the target task is a 4-categorical employment prediction. In this situation, $r^{\text{Opt}}(\lambda)>1$ for some $0\le\lambda<1$. Please see Figures 3 (b), 4-6 (c), and the explanation therein for this new addition.
>
>
>
> > - Include proof sketches in the main text
>
> Thanks for your constructive suggestion. We included proof sketches of all theorems in the main text.
>
>
>
> > - Implementation details: describe how neural architectures for ARL were selected. Are the number of network parameters comparable to the number of kernel parameters for K-$\mathcal T_{\text{Opt}}$?
>
> We started with a linear network for the encoders of ARL and HSIC-IRepL and gradually increased the number of layers and neurons until over-fitting was observed. For RFF dimensionality ($d_{\text{RFF}}$), we started with a small value, and gradually increased it until reaching the maximum possible performance for $\lambda=0$ (i.e., the standard unconstrained representation learning) on the corresponding validation sets. For the toy example, the number of encoder network parameters for ARL and HSIC-IRepL methods is $735$. While that is $100=100 * r^{\text{Opt}}(\lambda\rightarrow 1)\le100 * r^{\text{Opt}}(\lambda)\le100 * r^{\text{Opt}}(\lambda=0)=1500$  for K-$\mathcal T_{\text{Opt}}$ ($d_{\text{RFF}}=100$). For the CelebA dataset, ARL and HSIC-IRepL have $41,024$ parameters in their encoder. While that is $1000=d_{\text{RFF}}*1$ for K-$\mathcal T_{\text{Opt}}$. For the Folktables dataset, the ARL and HSIC-IRepL have $15,616$ parameters. While that is $5000=d_{\text{RFF}}*1$ for K-$\mathcal T_{\text{Opt}}$. We added these implementation details to Section 6.5. Please see our changes in the revised manuscript.
>
>
>
> > - In addition to Figures 3-4, since Equation (11) characterizes the trade-off between Dep$(Z, Y)$ and Dep$(Z, S)$. It would be good to include this trade-off in Section 6.6.
>
> We appreciate your constructive suggestion. We included this trade-off. Please see Figure 5 in the revised manuscript.
>
>
>
> > - If possible, it would be good to calculate theoretical upper/lower  bounds for this experimental setup, and include them in Figures 3-4
>
> We are not sure whether we understand this comment correctly or not. Please note that theoretical upper/lower bounds on Dep$(Z, Y)$ and $\text{Dep}(Z, S)$ will not help with bounding accuracy and DPV (or KCC) since there is no analytical relation (to the best of our knowledge) between accuracy and Dep$(Z, Y)$, or between DPV (or KCC) and Dep$(Z, S)$.
>
>
>
> > - "reverse of a non-parametric measure" should be "negative of a non-parametric measure"
>
> Thanks for your careful attention. "reverse of a non-parametric measure" has been changed to "negative of a non-parametric measure".
>
>
>
> > - "performs as optimum as" should be "performs as optimally as"
>
> "performs as optimum as" has been changed to "performs as optimally as".
>
>
>
> > Broader Impact Concerns: No major concerns, but the authors could comment on the distinction between characterizing tradeoffs and designing models/datasets to obtain specific invariance/fairness properties.
>
> We added the "Broader Impact" Section after the "Conclusion" Section and we commented there that "even though the objective of this paper is to characterize the utility-invariance trade-off, as a byproduct, our paper proposes an algorithm that can be used to learn fair representations of data".

---

> > ### Comment · Reviewer_tLLR · 2022-11-03
> > **Update**
> >
> > - Thank you for clarifying the use of $r^{Opt}(\lambda)$. This was confusing because of the last sentence in **Embedding Dimensionality**: "... and therefore we let $r = 1$ for all methods and all $0 \leq \lambda <1$." The addition of Folktables-NY with $r^{Opt}(\lambda) > 1$ in Figures 3(b), 4-6(c) also makes this distinction clearer. Note that the caption for Figure 5 is not updated with Folktables-NY.
> >
> > - My suggestion for upper/lower bounds is to connect the empirical results in Section 6.6 to cited related work on similar utility-fairness tradeoffs, e.g. Zhao & Gordon (2019), Zhao et al. (2020). If the proposed exact tradeoff is a different setting, then the authors should explain why the revised Figure 5 is incomparable to existing bounds.

---

> > > ### Author Response · Authors · 2022-11-05
> > > **Reply to Update of Reviewer tLLR**
> > >
> > > > Thank you for clarifying the use of  $r^\text{Opt}(\lambda)$. This was confusing because of the last sentence in **Embedding Dimensionality** ...
> > >
> > > Thanks for your positive comments.
> > >
> > > > Note that the caption for Figure 5 is not updated with Folktables-NY.
> > >
> > > We appreciate your careful attention. We modified the manuscript to update the caption of Figure 5 with Folktables-NY.
> > >
> > > > My suggestion for upper/lower bounds is to connect the empirical results in Section 6.6 to cited related work on similar...
> > >
> > > ## First:
> > > We point out that there were two minor inaccuracies in comparing our work with Zhao et al. (2019a), Zhao and Gordon (2019), and Zhao et al. (2020) in  Section 2.3 under **Restricted Class of Attributes**:
> > >
> > > - The citation Zhao et al, (2019a) was swapped with Zhao & Gordon (2019). This swap is corrected in the new modified manuscript.
> > > - We did not mention that for the classification task in Zhao et a. (2020), $S$ and $Y$ are restricted to binary RVs. This restriction is evident from Definition 4.1, Theorem 4.1, Corollary 4.1, Corollary 4.2, and Proposition 4.2 in Zhao et a. (2020).
> > >
> > > Please see the new modified manuscript for the correction of these two inaccuracies.
> > >
> > > ## Second:
> > > Please note that the main purpose of our paper is to address the shortcoming (in terms of applicability and precision) of existing works on characterizing the utility-invariance trade-off, and therefore we selected our experimental setting accordingly. To the best of our knowledge, the only related works in the **utility-fairness trade-offs family** that are general enough (multi-class classification, regression, or combination of these two) to be applied to our experimental setting are
> > >
> > > 1. Sadeghi et al. (2019)
> > >
> > > 2. Zhao & Gordon (2019)
> > >
> > > For instance, in Zhao et al. (2020), both $S$ and $Y$ are restricted to be continuous or binary at the same time and it is not possible to have $Y$ binary while $S$ is continuous, i.e., this related work cannot be applied to the toy example, Folktables-WA, and Folktables-NY datasets). Also, it is not possible to have $Y$ binary while $S$ is categorical, i.e., this related work is inapplicable to the CelebA dataset as well.
> > >
> > > 1. We are already experimentally comparing with Sadeghi et al. (2019) as denoted by SARL in Figure 4 (a-d) and 7 (c) in Section 6.
> > >
> > > 2. The related work by Zhao & Gordon (2019) is restricted to categorical $Y$ and $S$ only and therefore this related work can be applied only to CelebA dataset since in our other experiments $S$ is continuous. Nevertheless, Zhao & Gordon (2019) has provided a lower bound on the summation of classification errors for each class of $S$ (not the overall classification error) only, under the total invariance scenario, i.e., when $Z$ is independent of $S$. Recall that in the CelebA dataset, $Y$ is binary and $S$ is $4$-class categorical. As a result on the CelebA dataset, this bound reads (see Theorem 12 in Zhao & Gordon (2019))
> > >
> > > $$
> > > \\sum_{i=1}^{4}\\text{Pr}[\\widehat Y\\not = Y| S=i]\\ge \\text{TV-Barycenter}, \\qquad (A)
> > > $$
> > >
> > > where the left-hand side of $(A)$ is the summation of the classification errors for each class of $S$ and TV-Barycenter is defined as the following optimization problem:
> > >
> > > $$
> > > \\text{TV-Barycenter}:\\qquad \\min_q \\sum_{i=1}^4|q-p_i| \\qquad \\text{ s.t.} \\qquad 0 \\le q \\le 1,
> > > $$
> > >
> > > where $p_i=\text{Pr}[Y=1| S=i]$ or $p_i=\text{Pr}[Y=0| S=i]$.
> > >
> > > For CelebA dataset, the left-hand side of (A) is
> > > $
> > > \\sum_{i=1}^{4}\\text{Pr}[\\widehat Y\\not = Y| S=i]=1.89 \\nonumber
> > > $
> > > and the right-hand side of (A) is
> > > $
> > > \\text{TV-Barycenter} =0.63 \\nonumber,
> > > $
> > > that is obtained optimally via linear programming.
> > >
> > > In addition to the fact that the bound in $(A)$ refers to only a single point ($Z$ independent of $S$) in utility-invariance trade-off, also, $\sum_{i=1}^{4}\text{Pr}[\widehat Y\not = Y| S=i]$ in $(A)$ ignores the marginal distribution of $S$ and therefore it is different than the overall error rate (i.e., $1-$accuracy) reported in Figure 4 of our paper, that is
> > > $$
> > > \\text{Error}=1-\\text{Accuracy}=\\text{Pr}[\widehat Y\\not = Y | = \\sum_{i=1}^{4}\\text{Pr}[\widehat Y\\not = Y | S=i] \\text{Pr}[S=i]\\nonumber.\\qquad (B)
> > > $$
> > > Hence, the discrepancy between $(A)$ and $(B)$,  makes the bound in $(A)$ incompatible with our paper's utility-invariance trade-off (Figure 4).

---

### Review · Reviewer_Web1 · 2022-10-20

**Summary Of Contributions:**

This paper investigates a general-purpose invariant representation learning setup (usable for domain adaptation, fairness applications, etc.) where the class of functions for the representation are restricted to an RKHS. The authors demonstrate that it is possible to derive Pareto-optimal models in closed form (where Pareto-optimality balances predictive performance versus invariance). The authors further derive rates of convergence.

**Audience:**

Yes

**Broader Impact Concerns:**

One of the primary motivations for this method is fair representation learning (in the form of demographic parity). Many who have studied algorithmic fairness have mentioned that this can be a flawed notion of fairness (e.g. https://arxiv.org/abs/1610.07524 and https://arxiv.org/abs/1610.02413). It is worth mentioning this fact in the paper.

**Claims And Evidence:**

Yes

**Requested Changes:**

# Critical changes

- The HSIC-IRL method is not mentioned in the related work. It seems highly related to your proposed method (is the only difference using Dep instead of HSIC?). Please explain how your method differs from HSIC-IRL.
- Explain how the `Dep` approximation to HSIC affects the guarantees of HSIC.
- Clarify how Theorem 6 implies the claims in Section 5.2

# Minor changes:

- You use the phrase "modes of dependence" frequently. However, I have no idea what this means; it is never defined.
- The convergence rate proofs require `f` to be bounded. However, this will not be the case generally if using a universal RKHS. Please comment about why this assumption is okay.
- It is worth mentioning in the paper that Dep, unlike HSIC and other metrics, is asymmetric.
- `\boldsymbol K_S` (below equation 9) is not defined.
- Theorem 3: what is `f` (not bolded)?
- Figure 2: Why do you use `Cov` and not `Dep` in the diagram?

**Strengths And Weaknesses:**

# Strengths

- The results in this paper are strong and non-trivial. It is very interesting to see that the RKHS formulation leads to closed form results.
- The method is very general and can be applied to lots of settings.
- The proofs in this paper are comprehensive and easy to follow.
- This topic (invariant representation learning) is very relevant.

# Weaknesses

Overall, I believe that the strengths of this paper outweighs its weaknesses. Nevertheless, here are several points that should be addressed by the authors

## Claims in the introduction are a bit too strong?

The paper claims that this paper derives "a closed-form solution for the global optima of the underlying optimization problem..." However, it should be stated that this closed-form optimum only holds for a non-standard objective function (Dep), and requires that the empirical risk function and the invariance constraint use the same function (Dep). While I don't think that this hurts the results of the paper, the authors may wish to tone down the language used in the introduction and abstract.

## Properties of Dep are not explored

The `Dep` function introduced by the authors is motivated to be a tractable version of the HSIC function. The authors discuss the desirable properties of HSIC (i.e. it is zero when the two functions are independent) but they do not discuss how the `Dep` approximation affects these properties. (This point is related to the first point: I think that the author's claims in the introduction are somewhat incomplete since the "exact tradeoff" is not totally exact - due to the Dep approximation - and the authors never quantify how much this approximation affects their guarantees.)

## How does Theorem 6 support the claims in Section 5.2?

In Section 5.2, the authors argue that optimizing the proposed Dep objective function can achieve the Bayes estimator, and demonstrate this with Theorem 6. I was not able to connect the dots between how Theorem 6 implies this claim. The Theorem states that "there exists a representation `\boldsymbol f^*`" that achieves the Bayes estimator; however, it does not state that optimizing the proposed objective function yields `\boldsymbol f^*`.

---

> ### Author Response · Authors · 2022-10-29
> **Reply to Reviewer Web1 - Part 1**
>
> > #### Claims in the introduction are a bit too strong?
>
> We modified the Abstract and Introduction. Please see our changes in blue in the revised manuscript.
> For instance, please see "Under the assumption that the distribution of a low-dimensional projection of high-
> dimensional data is approximately normal, we derive a closed-form solution for the global optima
> of the underlying optimization problem for encoders in RKHSs." in the Abstract.  We also changed the word "exact optimal trade-off" to "near-optimal trade-off" in the Abstract and Introduction.
>
>
>
> > - The HSIC-IRL method is not mentioned in the related work. It seems  highly related to your proposed method (is the only difference using Dep instead of HSIC?) Please explain how your method differs from  HSIC-IR
>
> We added the following comparison to HSIC-IRepL in related work that read as
> "HSIC, a universal measure of dependence, has been adopted by prior work (e.g., Quadrianto et al. (2019)) to quantify all types of dependencies between Z and S. However, these methods adopt stochastic gradient descent for optimizing the underlying non-convex optimization problem. As such, they fail to guarantee that the representation learning problem converges to a global optima. In contrast, we obtain a closed-form solution for the globally optimal encoder and its corresponding representation while detecting all modes of dependence between $Z$ and $S$."
>
>
>
> > - Explain how the Dep$(Z, S)$ approximation to HSIC affects the guarantees of HSIC.
>
> Below equation (8), we explained the Dep$(Z, S)$ guarantees on universality (i.e., like HSIC, Dep$(Z, S)=0$ iff $Z$ is independent of $S$):
> "The measure $\text{Dep}(Z, S)$  captures all modes of non-linear dependence under the assumption that the distribution of a low-dimensional projection of high-dimensional data is approximately normal (Diaconis & Freedman, 1984), (Hall & Li, 1993). To see why this reasoning is relevant, we note from (5) that $Z$ can be expressed as $Z=\boldsymbol{\Theta} V$, where $V\in \mathbb R^n$ and $\boldsymbol{\Theta} \in \mathbb R^{r\times n}$. This indicates that for large $n$ and small $r$ (which is the case for most real-world datasets), $Z$ is indeed a low-dimensional projection of high-dimensional data.  In other words, $\left(Z, \beta_S(S)\right)$ is approximately a jointly Gaussian RV. In our numerical experiments in Section 6, we empirically observe that $\text{Dep}(Z, S)$ enjoys a monotonic relation with the underlying invariance measures and captures all kinds of dependency in practice, especially as $Z$ independent of $S$."
> However, for the sake of completeness, we now added the following explanation right after the above paragraph:
> "Nevertheless, if the normality assumption on the distribution of $(Z,\beta_S(S))$ fails, Dep$(Z, S)$ reduces to measuring the linear dependency between $Z$ and $\beta_S(S)$ for all Borel functions $\beta_S$. This corresponds to measuring the mean independency of $Z$ from $S$, i.e., how much information a predictor (linear and non-linear) can infer (in the sense of MSE) about $Z$ from $S$. See Appendix H for more technical details on mean independence."
>
>
>
> > - Clarify how Theorem 6 implies the claims in Section 5.2
>
> We appreciate your constructive comment. The purpose of Theorem 6 is to indicate that the optimal encoder $\boldsymbol{f}^*$ resulted from our proposed loss function, i.e., $\text{Dep}\left(\boldsymbol{f}(X), Y\right)$ can achieve the Bayes estimation. We implicitly mentioned in Theorem 6 that $\boldsymbol {f}^*$ is optimized using Dep$(\boldsymbol{f}(X), Y)$ by stating that "Let $\boldsymbol{f}^*$ be the optimal encoder obtained by (14)", where (14) is the optimal encoder learned by Dep$(Z, Y)$.  For the sake of a clear connection between $\text{Dep}\left(\boldsymbol{f}(X), Y\right)$ and Bayes estimator, we now explicitly state in Theorem 6 that " Let $\boldsymbol{f}^*$ be the optimal encoder by maximizing Dep$(\boldsymbol{f}(X), Y)$". Please see Theorem 6 in the modified manuscript. It now reads
>
> **Theorem 6.** Let $\boldsymbol{f}^*$ be the optimal encoder by maximizing Dep$(\boldsymbol{f}(X), Y)$, where $\gamma\rightarrow 0$ and $\mathcal H_Y$ is a linear RKHS. Then, there exists a linear regressor on top of the representation $Z^*=\boldsymbol{f}^*(X)$ that can perform as optimally as the Bayes estimator:
> $$
> \\min_{\boldsymbol W\\in\mathbb R^{d_Y\\times r}, \\boldsymbol b\\in \\mathbb R^{d_Y}}\\mathbb E_{X, Y} \\left[\\|\\boldsymbol W Z^*+\\boldsymbol b-Y\\|^2\\right] =\\inf_{h \\text{ is Borel}}\\mathbb E_{X, Y} \\left[\\|h(X)-Y\\|^2\\right]=\\mathbb E_{X, Y} \\left[\\|\\mathbb E[Y|\\,X]-Y\\|^2\\right].
> $$

---

> > ### Comment · Reviewer_Web1 · 2022-11-03
> > **Small further clarification**
> >
> > Thank you for the follow up. Your clarifications/revisions adequately address my concerns.
> >
> > One small clarification:
> >
> > > *Theorem 6*: ...that can perform as optimally as the Bayes estimator.
> >
> > Why are you using the wording "that can perform as optimally as the Bayes estimator," rather than "such that $\boldsymbol W Z^*$ is the Bayes estimator?"

---

> > > ### Author Response · Authors · 2022-11-03
> > > **Reply to "Small further clarification",  Reviewer Web1**
> > >
> > > We appreciate your positive comments.
> > >
> > > > Why are you using the wording "that can perform as optimally as the Bayes estimator," ...
> > >
> > > Thanks for your constructive suggestion. For the sake of accessibility and clarity, we can modify Theorem 6 to **equivalently** read:
> > >
> > > **Theorem 6.** Let $\boldsymbol{f}^*$ be the optimal encoder by maximizing Dep$(\boldsymbol{f}(X), Y)$, where $\gamma\rightarrow 0$ and $\mathcal H_Y$ is a linear RKHS. Then, there exist  $\boldsymbol{W}\in \mathbb R^{ d_Y\times r}$ and $\boldsymbol{b}\in \mathbb R^{d_Y}$ such that $\boldsymbol{W} \boldsymbol{f}^*(X)+\boldsymbol{b}$ is the Bayes estimator, i.e.,
> > > $$
> > > \\mathbb E_{X, Y} \\left[\\|\\boldsymbol W \boldsymbol{f}^*(X)+\\boldsymbol b-Y\\|^2\\right] =\\inf_{h \\text{ is Borel}}\\mathbb E_{X, Y} \\left[\\|h(X)-Y\\|^2\\right]=\\mathbb E_{X, Y} \\left[\\|\\mathbb E[Y|\\,X]-Y\\|^2\\right].
> > > $$
> > >
> > > We note that the modification does not affect the proof of this Theorem.

---

> ### Author Response · Authors · 2022-10-29
> **Reply to Reviewer Web1 - Part 2**
>
> > - You use the phrase "modes of dependence" frequently. However, I have no idea what this means; it is never defined.
>
> By "all modes of dependence" we mean all types of linear and non-linear relations in contrast to only linear or monotonic relations. We added this definition to the footnote of page 2.
>
>
>
> > - The convergence rate proofs require $f$ to be bounded.  However, this will not be the case generally if using a universal RKHS.  Please comment about why this assumption is okay.
>
> Please note that for any  $\boldsymbol{x}$  in the training set, $f_j(\boldsymbol x)$ can be obtained as $f_j(\boldsymbol {x})=\sum_{i=1}^n \theta_{ji} k_X(\boldsymbol{x}_i, \boldsymbol{x})$. We can assume that $k_X(\cdot, \cdot)$ is bounded. For example, in RBF Gaussian and Laplacian RKHSs (that are universal), $k_X(\cdot, \cdot)\le 1$. This implies that $f^2_j(\boldsymbol x)\le \sqrt{n}\\|\boldsymbol{\theta}_j\\|$, where $\boldsymbol{\theta}_j$ is $j$-th row of $\boldsymbol{\Theta}$ in equation (5). One always can normalize $f_j(\boldsymbol x)$ by dividing it by the maximum of $\sqrt{n}\\|\boldsymbol{\theta}_j\\|$ over $j$s, or by dividing by the maximum  of $|f_j(\boldsymbol {x}_i)|$ over $i$s and $j$s. Notice that this normalization is only a scalar multiplication and has no effect on the invariance of $Z=\boldsymbol{f}(X)$ to $S$ and the utility of any downstream target task predictor $g_Y(Z)$. We added this explanation after Theorem 5.
>
>
>
> > - It is worth mentioning in the paper that Dep, unlike HSIC and other metrics, is asymmetric.
>
> We added the following statement after equation (7). "We note that Dep$(\cdot, \cdot)$, unlike HSIC and other kernelization-based dependence measures, is not symmetric. However, symmetry is not necessary for measuring statistical dependence."
>
>
>
> > $\boldsymbol{K}_S$ (below equation 9) is not defined.
>
> Please note that $\boldsymbol K_S$ is defined at the beginning of Lemma 1. For the sake of comprehension, we are now recalling the definition of $\boldsymbol K_S$ below equation (9).
>
>
>
> >Theorem 3: what is $f$ (not bolded)?
>
> We appreciate your careful attention. Now, $f$ is bolded ($\boldsymbol f$).
>
>
>
> >Figure 2: Why do you use $\text{Cov}$ and not $\text{Dep}$ in the diagram?
>
> Thanks for your careful attention again. Now, $\text{Dep}$ together with its definition is used in Figure 2.
>
>
>
> > Broader Impact Concerns: One of the primary motivations for this method is fair representation learning (in the form of demographic parity). Many who have studied algorithmic fairness have mentioned that this can be a flawed notion of fairness ...
>
> We added the "Broader Impact" Section after the "Conclusion" Section where we mentioned that "demographic parity can be an unsuitable fairness criterion in some practical scenarios (Hardt et al., 2016; Chouldechova, 2017) and some other fairness criterion like equalized odds (EO) or equality of opportunity (EOO) (Hardt et al., 2016) should be considered. Our method in this paper can be extended to other notions of fairness such as EO and EOO by modifying Dep(Z, S) to capture the dependency between Z and S, given Y. We leave this extension to future work."

---

### Author Response · Authors · 2022-12-20
**Camera Ready**

Thank you to all reviewers for their constructive feedback that helped us improve the paper's quality and clarity. We have now uploaded the camera-ready version of our paper.